# Sub-picosecond thermalization dynamics in condensation of strongly coupled lattice plasmons

Aaro I. Väkeväinen[1], Antti J. Moilanen [1], Marek Nečada[1], Tommi K. Hakala[2], Konstantinos S. Daskalakis [1] & Päivi Törmä [1]✉

Bosonic condensates offer exciting prospects for studies of non-equilibrium quantum dynamics. Understanding the dynamics is particularly challenging in the sub-picosecond timescales typical for room temperature luminous driven-dissipative condensates. Here we combine a lattice of plasmonic nanoparticles with dye molecule solution at the strong coupling regime, and pump the molecules optically. The emitted light reveals three distinct regimes: one-dimensional lasing, incomplete stimulated thermalization, and two-dimensional multimode condensation. The condensate is achieved by matching the thermalization rate with the lattice size and occurs only for pump pulse durations below a critical value. Our results give access to control and monitoring of thermalization processes and condensate formation at sub-picosecond timescale.

[1] Department of Applied Physics, Aalto University School of Science, P.O. Box 15100, Aalto FI-00076, Finland. [2] Institute of Photonics, University of Eastern Finland, P.O. Box 111, Joensuu FI-80101, Finland. ✉email: paivi.torma@aalto.fi

Ideal Bose–Einstein condensation (BEC) means accumulation of macroscopic population to a single ground state in an equilibrium system, with emergence of long-range order. Bosonic condensation in non- or quasi-equilibrium and driven-dissipative systems extends this concept and offers plentiful new phenomena, such as loss of algebraically decaying phase order[1,2], generalized BEC into multiple states[3], rich phase diagrams of lasing, condensation and superradiance phenomena[4,5], and quantum simulation of the XY model that is at the heart of many optimization problems[6]. Such condensates may also be a powerful system to explore dynamical quantum phase transitions[7]. Each presently available condensate system offers different advantages and limitations concerning studies of non- and quasi-equilibrium dynamics. The ability to tune interactions precisely over a wide range is the major advantage of ultracold gases[8,9]. Polariton condensates[10–18] offer high critical temperatures compared to ultracold gases. In photon condensates[19,20], the thermal bath is easily controlled[21,22]. Recently periodic two-dimensional (2D) arrays of metal nanoparticles, so-called plasmonic lattices or crystals[23], have emerged as a multifaceted platform for room temperature lasing and condensation at weak[24–28] and strong coupling[29–31] regimes.

In plasmonic lattices, the lattice geometry and periodicity, the size and shape of the nanoparticles, and the overall size of the lattice can be controlled with nanometer accuracy and independent of one another. The energy where condensation or lasing occurs is given by the band edge energy that depends on the period of the array. Remarkably the band edge energy and the dispersion are extremely constant over large lattices (accuracy 0.1%[28]). In semiconductor polariton condensates, disorder in the samples often leads to traps and fragmentation[15], or condensates may be trapped by geometry[16,19]. Thus plasmonic lattices offer a feature complementary to other condensate systems, namely that propagation of excitations over the lattice can be used for monitoring time-evolution of such processes as thermalization: each position in the array can be related to time via the group velocity, and there are no spurious effects due to non-uniformity of the sample. Spatially resolved luminescence was utilized in this way in the first observation of a BEC in a plasmonic lattice[28].

Here, we show that formation of a condensate with a pronounced thermal distribution is possible at a 200 fs timescale and attribute this strikingly fast thermalization to partially coherent dynamics due to stimulated processes and strong coupling. We observe a unique double-threshold phenomenon where one-dimensional (1D) lasing occurs for lower pump fluences and 2D multimode condensation, associated with thermalization, at higher fluences. The transition between lasing and condensation shown in our work is different from previous condensates[16,28,32–36]: it relies on matching the system size, propagation of excitations, and the thermalization dynamics. Importantly we find a peculiar intermediate regime showing features of a thermalization process but no macroscopic population at the lowest energy states. This regime allows us to reveal the stimulated nature of the thermalization process by the behavior of the luminescence in lattices of different sizes. As a direct evidence of the ultrafast character of the thermalization and condensation process, we show that it occurs only for pump pulse durations below a critical value of 100–250 fs. In the following, we first present characterization, such as luminescence spectra and spatial coherence, of the lasing and condensation phenomena and then focus on the main results: the stimulated nature of the thermalization process and the dramatic effect of the pump pulse duration.

## Results

**System.** Our system consists of cylindrical gold nanoparticles in a rectangular lattice overlaid with a solution of organic dye molecule IR-792 (see details in the Section "Methods: Samples"). The lattice supports dispersive modes, so-called surface lattice resonances (SLRs), which are hybrid modes composed of localized surface plasmon resonances at the nanoparticles and the diffracted orders of the periodic structure[23,37]. The electric field of the SLR modes is confined to the lattice plane in which the SLR excitations can propagate. An SLR excitation can be considered a bosonic quasiparticle that consists (mostly) of a photon and of collective electron oscillation in individual metal particles.

The SLR modes are classified to transverse magnetic (TM) or transverse electric (TE) depending on the polarization and propagation direction, as defined in Fig. 1b. The measured dispersions are displayed in Fig. 1c–e. In the presence of dye molecules, the SLR dispersion shifts downwards in energy with respect to the initial diffracted order crossing. Moreover the TE modes begin to bend when approaching the molecular absorption line at 1.53 eV. These observations indicate strong coupling between the SLR and molecular excitations[38]; a coupled modes fit to the data gives a Rabi splitting of 164 meV (larger than the average line width of the molecule (150 meV) and SLR (10 meV)) and an exciton part of 23% at $k = 0$. In the following, we refer to the hybrids of the SLR mode excitations and molecular excitons as polaritons, for brevity. The coherence length of the polaritons (samples that are not pumped) is 24 μm, as obtained from the observed dispersions. Here, we have a high molecule concentration in a liquid gain solution, in contrast to previous studies[26,29]. The plasmonic lattice, optimized for creating the condensate, has a particle diameter of 100 nm and height of 50 nm, the period in $y$- and $x$-direction of $p_y = 570$ nm and $p_x = 620$ nm, dye concentration of 80 mM, and a lattice size of $100 \times 100$ μm². The period $p_y$ is varied between 520 and 590 nm, and the lattice size between $40 \times 40$ and $200 \times 200$ μm².

We excite the sample with laser pulses at 1 kHz repetition rate and central wavelength of 800 nm, and resolve the luminescence spectrally as a function of angle and spatial position on the array, Fig. 1a (for details see Section "Methods: Transmission, reflection, and luminescence measurement setup"). The pump does not directly couple to the SLR modes, and only a small fraction of the photons are coupled to the single particle resonance and/or absorbed by molecules within a near vicinity of the nanoparticle lattice. Active region of the dye molecules lies within a few hundred nanometers from the lattice plane, shown experimentally in refs. [27,39–41], and molecules further away are unlikely to couple to the SLR modes.

**Transition as a function of pump fluence.** We study the luminescence properties of the plasmonic lattice as a function of pump fluence, that is, the energy per unit area per excitation pulse, and find a prominent double-threshold behavior. The system is excited with an $x$-polarized 50 fs laser pulse that has a flat intensity profile and a size larger than the lattice. Excited polariton modes continuously leak through radiative loss, and therefore the observed luminescence intensity is directly proportional to the population of the polaritons. We record real space and momentum ($k$-)space intensity distributions and the corresponding spectra, the photon energy is $E = hc/\lambda_0$ and the in-plane wave vector $k_{x,y} = 2\pi/\lambda_0 \sin(\theta_{x,y})$.

The sample luminescence as a function of pump fluence is presented in Fig. 2. The total luminescence intensity reveals two non-linear thresholds and a linear intermediate regime, shown in Fig. 2a. The line spectra, shown as insets, are obtained by integrating the real space spectra along the $y$-axis and unveil the

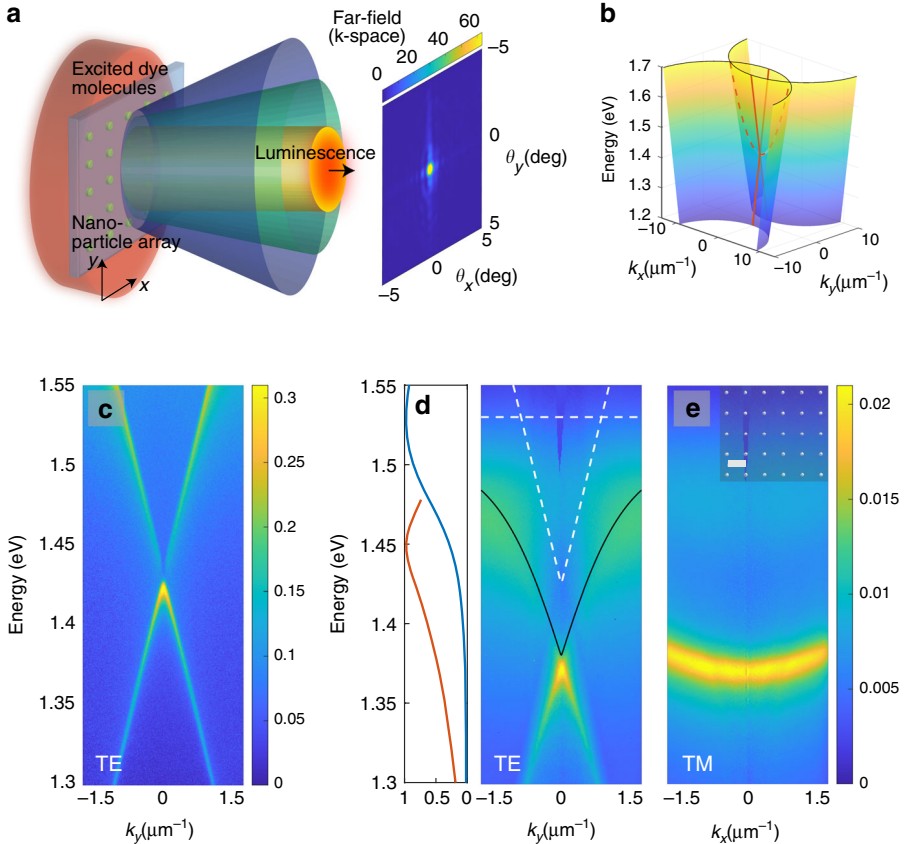

**Fig. 1 Schematic of the system and dispersion of the modes. a** Artistic illustration of the experimental configuration. **b** Light cones for the diffracted orders $(0, -1)$ and $(0, +1)$ that arise for $x$-polarized nanoparticles. Crosscut along $k_y$ ($k_x = 0$) is called the TE mode (red solid lines) and along $k_x$ ($k_y = 0$) the TM mode (red dashed line). For the TE mode, the polarization is perpendicular to the propagation direction ($\mathbf{e}_x, k_y$), and for the TM mode parallel ($\mathbf{e}_x, k_x$). Crosscuts of the SLR dispersions are experimentally obtained by measuring the transmission (**c**) without the molecule or (**d, e**) reflection with 80 mM solution of IR-792. In **d** the SLR dispersion exhibits a red shift and an avoided crossing with the absorption transition of the molecule. The absorption line and the uncoupled SLR mode are depicted with white dashed lines and the lower-polariton branch given by a coupled-modes-model fit is shown with a black line. Absorption and emission spectra of IR-792 are displayed in the left panel with blue and red lines, respectively. Scanning electron micrograph of the nanoparticle array is shown as an inset in **e**, the scale bar is 500 nm.

population of polaritons as a function of energy. At the first threshold, Fig. 2b, c, lasing (or polariton lasing/polariton condensation) typical for nanoparticle arrays[24,27,29,30] is observed throughout the array. Increasing the pump fluence beyond the first threshold, Fig. 2d, e, the luminescence becomes more intense in the central part compared to the top and bottom parts of the array. Moreover luminescence at the center takes place at a lower energy than at regions closer to the edges. We interpret this red shift as a signature of polariton population undergoing a thermalization process and propagating along the array in $+y$ and $-y$ directions, discussed below. At the second threshold, Fig. 2f, g, the system undergoes a transition into a condensate (that presents a Maxwell–Boltzmann (MB) distribution at higher energies): the real space intensity distribution shows uniform luminescence in the central part of the array, and the line spectrum (Fig. 2a, top inset) has a narrow peak at the band edge and a long thermalized tail at higher energies. A fit of the tail to the MB distribution (dashed line) gives the room temperature, $T = 313 \pm 2$ K (for more information see Section "Methods: Fits to the MB distribution"). We use the existence of a long (ranging over several $k_B T$, where $k_B$ is the Boltzmann constant) tail that fits the MB distribution as the criterion for distinguishing between what we call the condensate and lasing regimes. By the term condensate we thus refer here to the existence of a MB tail in addition to narrow-peaked population at low energy; peaked

population alone, without the tail, is referred to as lasing. Since we are at the strong coupling regime, the lasing that we see is actually polariton lasing which in the literature is called also polariton condensation[16,42]; we refer to this just by the words lasing regime, for brevity. As will be shown below, also the momentum-space confinement and spatial coherence properties of our condensate and lasing regimes differ dramatically, further confirming that the two phenomena are distinct.

The spectrometer counts per emitted condensate pulse correspond to a photon number of ~$10^9$ (see Section "Methods: Estimation of photon number in the condensate"), which is roughly $10^5$ times more than in the first BEC in a plasmonic lattice[28]. The increased luminescence is attributed to stimulated processes and differences in the sample as well as the pump and detection geometry (Supplementary Note 2). This tremendous improvement has increased the signal-to-noise ratio so that we can now observe a prominent thermal tail (it is likely to be even longer but the data is cut due to filtering out the pump pulse) which is an important signature of the efficiency of the thermalization process even in ultrafast timescales. The upgrade of luminescence intensity is also crucial for future fundamental studies and applications of this type of condensate. For instance, thermodynamic quantities can be determined using the observed photon distribution[43]. To produce a condensate, the periodicity must be tuned with

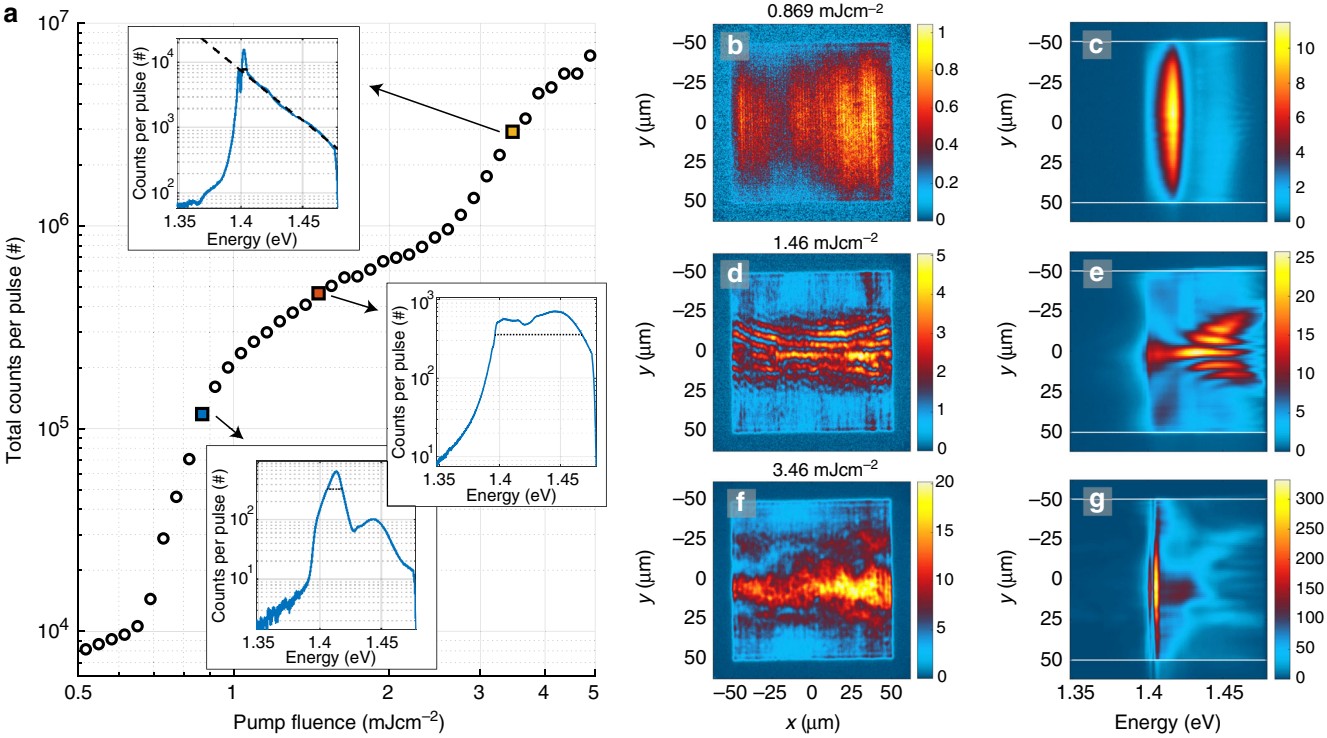

**Fig. 2 Pump fluence dependence, real space images and spectra. a** Double-threshold curve of the pump fluence dependence of the total luminescence intensity. Insets: Line spectra obtained by integrating over the real space spectra in the $y$-direction (between the white lines). The FWHM of the spectral peaks is marked in the insets: 12, 72, and 4.0 meV with increasing pump fluence. The dashed line in the top inset is a fit of the tail of the distribution to the Maxwell–Boltzmann distribution which gives a temperature of $313 \pm 2$ K (95% confidence bounds). **b–g** Left column: Real space images of the plasmonic lattice. Right column: Spectral information of the luminescence as a function of $y$-position. The pump fluence is **b**, **c** 0.87 mJ cm$^{-2}$, **d**, **e** 1.5 mJ cm$^{-2}$, and **f**, **g** 3.5 mJ cm$^{-2}$. The wave fronts in **d**, **e** arise from standing waves related to the momentum of the modes, for more information see Supplementary Fig. 1 and Note 1. The real space images are recorded for single pump pulses but the corresponding spectra are integrated over 500 (**c**, **e**) and 70 pulses (**g**). The results for the full range of pump fluences are presented in Supplementary Movie 1.

respect to the thermalization rate and the array size (Supplementary Note 1).

Three distinct regimes are also observed in the $k$-space intensity distributions. Figure 3 presents the $k$-space images and spectra for the same sample and pump parameters as in Fig. 2. Figure 3a–c shows that lasing spreads in the TM mode to large $k$ (i.e., large emission angles), whereas thermalization of the polaritons occurs mostly along the TE mode, see Fig. 3d–f. At the condensation threshold, Fig. 3g–i, we observe confinement in both $k_x$ and $k_y$, implying that the condensate has a 2D nature, in contrast to the lasing regime where confinement is observed only in $k_y$. Figure 3j shows line spectra obtained by integrating along $k_y$ from TE mode crosscuts (Fig. 3c, f, i); the spectrometer slit width of 500 μm corresponds to $\pm 1.3°$ around $\theta_x = 0$ in the 2D $k$-space images. Intriguingly, at the condensation threshold, we see multiple modes highly occupied at $k_y = 0$. The line spectrum at 3.49 mJ cm$^{-2}$ shows three narrow peaks followed by a thermalized tail. The full-width at half-maximum (FWHM) of the highest peak is 3.3 meV, significantly narrower than the bare SLR mode (10 meV). The spacing of the multiple peaks decreases toward lower energy, ruling out the possibility of a trivial Fabry–Pérot interference. We have also observed that the spacing does not depend on the periodicity or lattice size in a straighforward manner, and is not caused by waveguide effects[25,44]. Based on T-matrix simulations, we find that the cylindrical shape and finite size of the nanoparticles leads to three distinct modes around the $\Gamma$-point ($k_y = k_x = 0$) energy in an infinite lattice, one of these modes being highly degenerate. This highlights the role of the nanoparticles beyond providing a

periodic structure. The degeneracy is lifted by a finite lattice size, producing further distinct modes. The lattice size thus provides an additional means to tune the mode structure. For more information, see Supplementary Fig. 2 and Note 3. While at thermal equilibrium condensation occurs to the lowest energy, in driven-dissipative systems condensation to several modes at distinct energies is possible[3,5]. Since we observe a temporally integrated signal, we cannot rule out the possibility of a single-mode condensate temporally evolving between different states in the sub-picosecond scale.

Lasing and condensation transitions are expected to result in increased spatial and temporal coherence of the emitted light. Increase in temporal coherence was evidenced as the narrowing of spectral line width (Fig. 3j). To study the spatial coherence, we have performed a Michelson interferometer experiment as a function of pump fluence. In the Michelson interferometer, the real space image is split into two, one image is inverted and combined with the other one at the camera pixel array. The contrast of the observed interference fringes is extracted with a Fourier analysis of the spatial frequencies in the interfered image to exclude any artifacts produced by intensity variations in a single non-interfered image (Supplementary Figs. 3 and 4 and Note 4).

In Fig. 4, the fringe contrast (proportional to the first-order correlation function $g^{(1)}$) is shown as a function of pump fluence in both $y$- and $x$-directions of the lattice. High spatial coherence occurs in the $y$-direction over the array in both the lasing and condensation regimes (Fig. 4b, c). At the intermediate regime, the spatial coherence decreases, in agreement with the observation

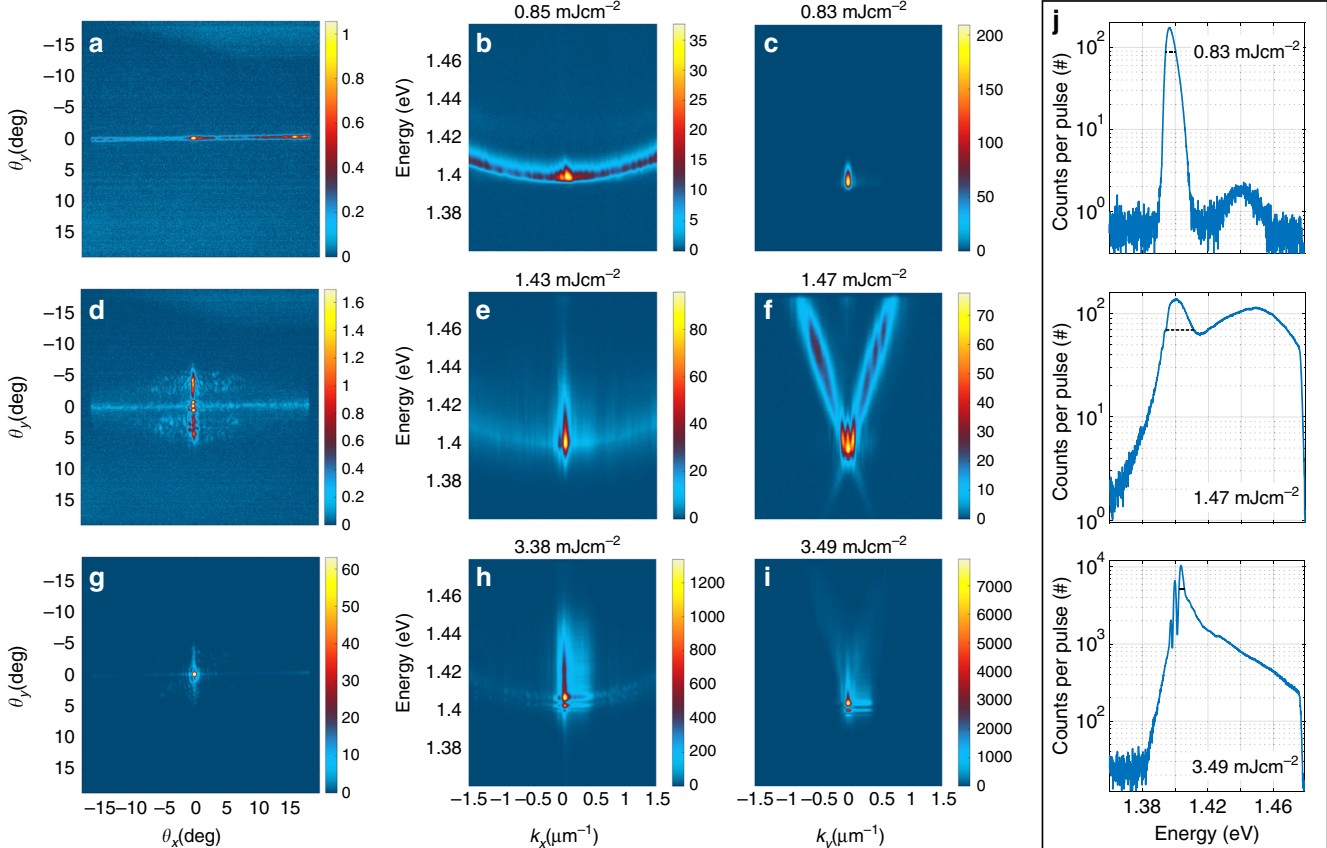

**Fig. 3 Momentum space images and spectra.** First column: 2D momentum (*k*-)space images. Second and third column: *k*-space spectrum in the TM and TE mode directions, respectively. The spectra of the TM and TE modes correspond to horizontal and vertical slices of the 2D *k*-space spectrum, respectively. The pump fluence is **a–c** 0.85 mJ cm$^{-2}$, **d–f** 1.5 mJ cm$^{-2}$, **g–i** 3.5 mJ cm$^{-2}$, as in Fig. 2. The images and the corresponding line spectra are integrated over 500 (**c**), 330 (**f**), and 20 pulses (**i**). The horizontal stretching of the emission peaks in **h**, **i** are CCD blooming artifacts. **j** Population distribution in the TE mode integrated from (**c**, **f**, **i**) along $k_y$. FWHMs are indicated with dashed black lines: 6.3, 17, and 3.3 meV with increasing pump fluence. The results for the full range of pump fluences are presented in Supplementary Movie 2.

that the thermalizing population dominates the luminescence signal. In the *x*-direction, spatial coherence is lower than in the *y*-direction over the whole pump range. However, the condensate clearly exhibits high spatial coherence also in *x*-direction, in contrast to lasing, which shows separated emission stripes in the real space image (see Fig. 4d, Supplementary Fig. 3). The spatial coherence measurements are in line with the observations from the 2D *k*-space images (Fig. 3), where (1) lasing exhibits confined luminescence along $k_y$ (the direction of feedback) but spreads along $k_x$, (2) condensation shows 2D *k*-space confinement. The coherence both in the lasing and condensation cases extends over the whole array (100 μm × 100 μm), thus the coherence length is at least four times larger than that of the samples without pumping (24 μm). In a future study, larger samples should be used for finding how the coherence decays (e.g., exponential and polynomial). Whether algebraically decaying phase order exists in 2D driven-dissipative systems is a subtle question[1,2,45,46].

The lasing and condensation take place at energies 1.397 and 1.403 eV, respectively, lower than the band-edge energy of the system without molecules, 1.423 eV. However the energies are blue-shifted from the lower polariton energy 1.373 eV (band edge in reflection) or 1.382 eV (fitted lower polariton branch; Fig. 1d), which are experimentally obtained by a reflection measurement and coupled modes model[38]. Since the whole dispersion gradually blue shifts as a function of pump fluence, it may be associated to degradation of strong coupling instead of originating from Coulomb interactions. Such saturation-caused non-linearity can

also be considered as effective polariton–polariton interaction[15] (a rough estimate of the strength of such interaction in our case is given in Supplementary Note 8). Based on the band-edge locations (1.423−1.403) eV/(1.423−1.373) eV, the coupling in the condensation regime has decreased to ~40% of the case without pumping. Note that the observed double-threshold behavior is different from semiconductor microcavity polariton condensates where condensation has a lower threshold than photonic lasing associated with loss of strong coupling[16,32].

**Stimulated nature of the thermalization.** At the intermediate regime, we observed red shift of the luminescence as a function of distance in *y*-direction, Fig. 2e. The trails of the red shift begin from the emission maximum of the dye molecule (~1.46 eV), at a certain distance from the array edges, and reach the band-edge energy (~1.40 eV) exactly at the center of the 100 × 100 μm$^2$ array.

To understand the red shift, we have recorded real space images and spatially resolved spectra for different lattice sizes at intermediate pump fluences sufficient to trigger the thermalization process, Fig. 5a–f. In a large array (Fig. 5e, f), we observe the trails of the red shift toward the center of the array, similarly as in the 100 × 100 μm$^2$ array (Fig. 2d, e), but the red-shifting populations do not merge at the center. In a small array (Fig. 5a, b), the situation looks different at first glance since the red shift seems to occur from the center of the array toward the edges.

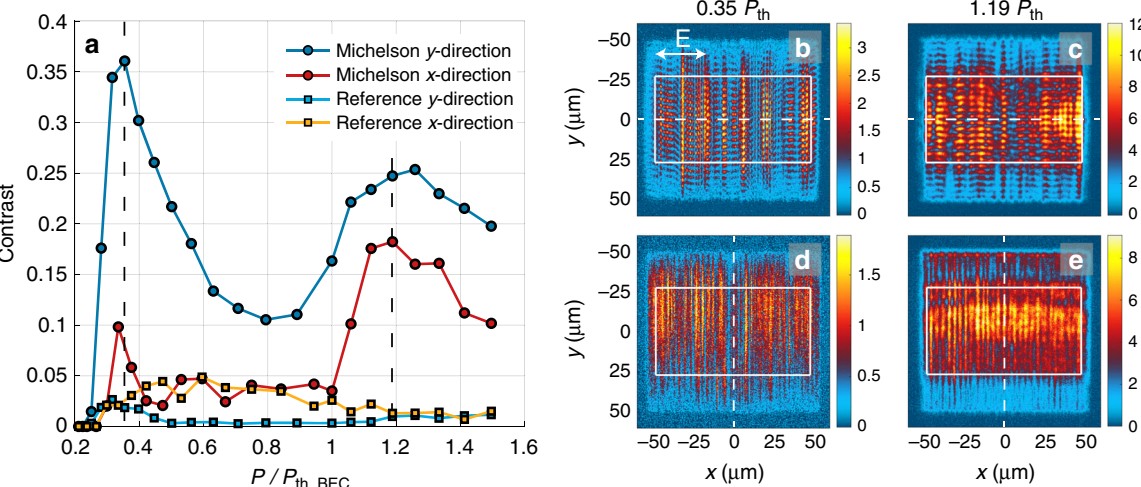

**Fig. 4 Spatial coherence measurement along y- and x-axis of the plasmonic lattice. a** The Michelson interferometer fringe contrast averaged over the region of interest (white boxes in **b–e**) as a function of pump fluence. For reference, the contrast is extracted also from non-interfered real space images, as light blue and yellow curves. **b–e** Interfered real space images, where one of the images is inverted with respect to the white dashed line, to obtain the spatial coherence between $-y$ and $+y$ positions (**b**, **c**) or $-x$ and $+x$ positions (**d**, **e**). The interference patterns are shown for pump fluences corresponding to the lasing and condensation regimes (dashed lines in **a**). To obtain the spatial coherence between $-x$ and $+x$ positions, instead of rotating the detection optics, we rotate the sample (the lattice) and pump polarization by 90°. The vertical stripes in **b** and **d** are due to one-dimensional lasing in y-direction, while the horizontal stripes (fringes) in **b**, **c** and vertical in **e** arise from spatial coherence in the overlapped images in the Michelson interferometer. For more details see Methods (Spatial coherence measurements) and Supplementary Note 4. The real space images are recorded for single pump pulses.

However, careful comparison of the distance between the array edge and the location where the red shift begins (see Supplementary Fig. 10 for details) reveals that for all arrays, for the given pump fluence, the distance is the same (~25 μm for 2.2 mJ cm$^{-2}$ pump fluence presented in Fig. 5a, c, e).

We explain this distance by stimulated emission pulse build-up time: the time between the maxima of the population inversion and the output pulse as defined in the rate-equation simulation in Fig. 5g (see[39] and Supplementary Note 5 for description of the model; note that we use this model just to illustrate the concept of pulse build-up in the thermalization process, not to describe the condensate or lasing observations). Pulse build-up is a well-known phenomenon in Q-switched lasers[47]. In our system, the pump pulse excites the molecules, and the polaritons begin to propagate when the first photons populate the modes. The modes then gather gain while propagating, and therefore the peak of the stimulated emission pulse appears after a certain distance traveled along the array. This distance is seen as the dark zones in real space measurements, and it corresponds to the pulse build-up time. Note that this pulse is different to a lasing or condensate pulse since the line width of the luminescence is large and spatial coherence is small. By summing up such spatial intensity profiles of the thermalizing pulses at every point along the lattice (Fig. 5h), we can reconstruct the real space intensity distributions (insets of Fig. 5a, c, e). The dark zones appear because the edges do not receive propagating excitations from outside the lattice; the dark zones at the edges have approximately half the intensity of the central part. In the small lattice, intensity at the edges is similar to that of the larger lattices but in the center it is only half of that. Moreover the wavy interference patterns in the central part (Fig. 5c, e) indeed only appear for arrays larger than $40 \times 40$ μm², where there are counter-propagating pulses that can interfere.

We found that the width of the dark zones depends on pump fluence as predicted by the rate-equation simulation and the theory of Q-switched lasers, namely the build-up time is inversely proportional to the pump fluence. Fig. 5i shows that the dark-

zone width follows the inverse of the pump fluence until it saturates at around 3 mJ cm$^{-2}$ (corresponding to the BEC threshold) to a value below 20 μm (~100–140 fs). The inset in Fig. 5i shows the pulse build-up time extracted from our rate-equation simulation, and it displays a similar ~1/P dependence.

We attribute the red shift to a thermalization process. At the intermediate regime of pump fluences, however, a thermal distribution is not reached before the population decays. For higher fluences, a condensate peak and a tail with MB distributed population emerges. A classical thermal MB distribution in logarithmic scale is a straight line. In contrast, a peaked feature at low energies, together with a MB tail, is a characteristic of the BE distribution. At low energies ($E - \mu < k_B T$), the BE distribution can be approximated as $k_B T/(E - \mu)$. A distribution of this form appears in so-called classical condensation of waves (or Rayleigh–Jeans (RJ) condensation), resulting for instance from an interplay between random noise and suitable gain/loss profiles of an optical system[48–52]. Our system does not have the conditions typical for classical condensation and, most importantly, the observed linear-in-log-scale tail does not match a distribution of the form $k_B T/(E - \mu)$. To rule out the RJ condensation by the distribution, one needs to observe it for energies $E$ that are larger than the condensate peak energy by more than $k_B T$, because for $E - \mu < k_B T$ the BE and RJ distributions coincide. It is thus essential that we resolve the tail up to energies that are 75 meV above the condensate energy—three times larger than the room temperature $k_B T = 25$ meV.

As the observed distribution does not match with either classical MB or RJ distribution, one can ask how does a distribution resembling the equilibrium BE case form in a non-steady-state, driven-dissipative system. In the weak coupling regime, the answer is known. Our system is similar to the photon condensates[19,33,53] in the sense that dye molecules with a vibrational level structure provide the thermalization mechanism. Differences to the photon condensates are our type of excitations (plasmonic-photonic lattice modes), ultrafast time-scales, and strong light-matter coupling. It has been shown both

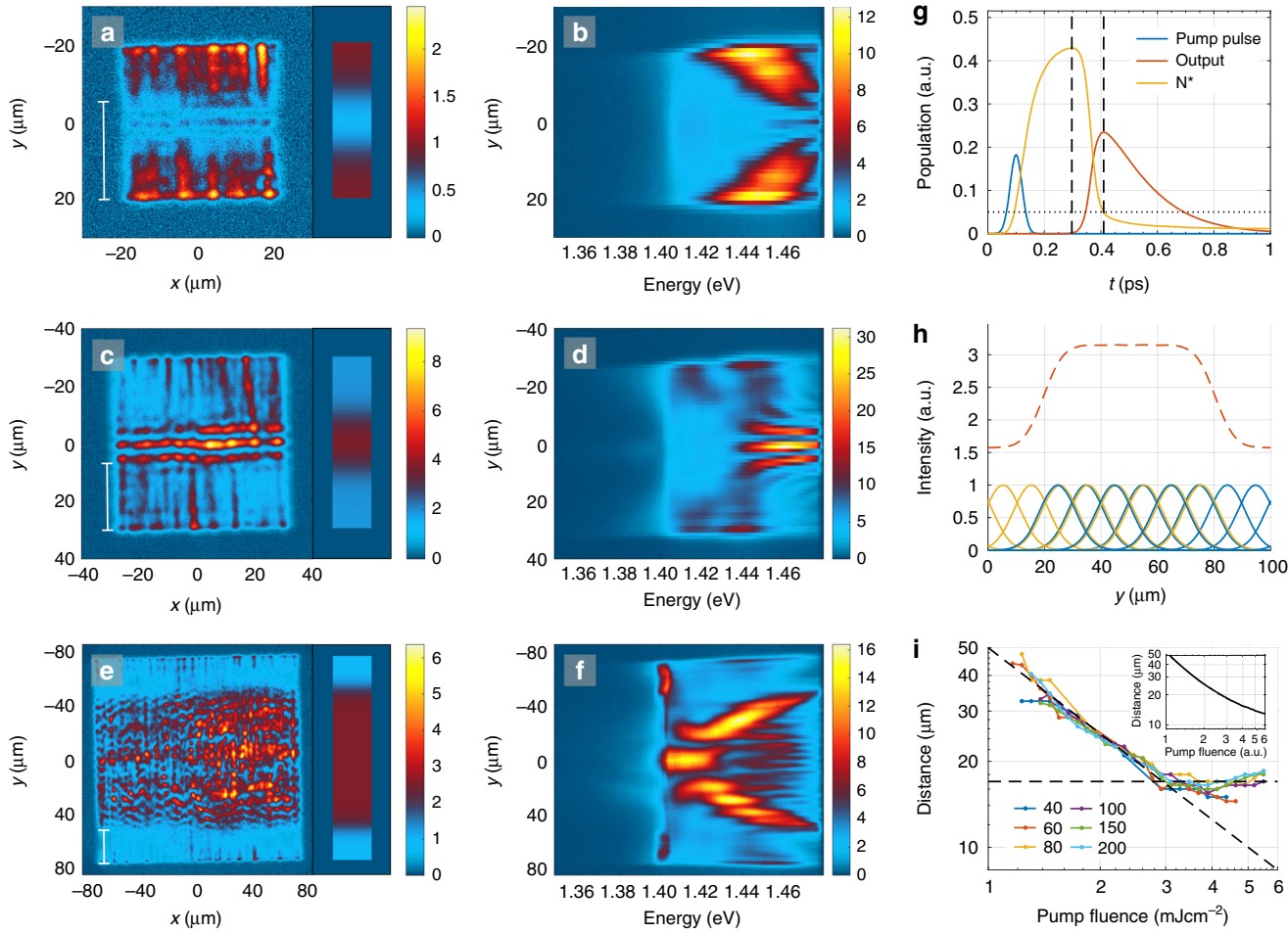

**Fig. 5 Observation of stimulated emission pulse build-up in finite size lattices. a–f** Real space images (left column) and spectra (middle column) at intermediate pump fluence (2.2 mJ cm$^{-2}$) for lattice sizes **a, b** 40 × 40 μm$^2$, **c, d** 60 × 60 μm$^2$, **e, f** 150 × 150 μm$^2$. Please note the different color scales in different panels. Line spectra for the different lattice sizes are presented in Supplementary Fig. 11. **g** Rate-equation simulation of stimulated emission. The pulse build-up time relevant for our system is marked by vertical dashed lines (the peaks of the population inversion $N^*$ and of the output pulse). Horizontal dotted line indicates the value at which $N^*$ overcomes the losses in the simulation. **h** Illustration of the sum (red dashed line) of spatial intensity profiles of the thermalizing pulses propagating to left (yellow) and right (blue) starting from everywhere along the y-axis of the array. The Gaussian shape approximates the increasing and decreasing intensity of the excitations as they gather gain and suffer losses under propagation. The results of such sum for different array sizes are shown as insets in **a, c, e** with the same false color as the real space images. **i** Measured distance from the array edge to the location where the red shift begins (indicated by the 25 μm scale bars in **a, c, e**) as a function of pump fluence for different lattice sizes. The legend indicates the square-array sizes in μm. The diagonal dashed line is the inverse of the pump fluence (50/P), and the horizontal dashed line indicates the saturation value (~18 μm). The inset shows the pulse build-up time (converted to distance by multiplying with the group velocity of the SLR mode) obtained from the rate-equation simulation.

experimentally and theoretically that, in the weak coupling regime, recurrent absorption and emission processes of light with molecules whose vibrational manifold is coupled to an external bath lead to thermalization and condensation with a BE(-type) distribution. This occurs both for continuous wave[19,33,36,53] and pulsed[28,54,55] pumping. The vibrational manifold serves as the energy loss channel to move the photon population towards lower energies, and thermal population of the vibrational states provides the temperature for the BE distribution. Due to the vibrational energy loss, the molecules may emit at lower energy than they absorb: this provides an effective coupling between photons at different energies, thus the thermalization process does not need scattering originating from Coulomb interaction as in inorganic semiconductor polariton condensation. The thermalization requires that several[52,56] (or just one[20]) absorption-emission cycles take place within the lifetime of the system. The speed of the thermalization process in general depends on the number of molecules, strength of the light–matter coupling, and

the number of photons since stimulated processes may be important. It is plausible that this mechanism or its modified form provides the thermalization also in our present strongly coupled case. Although the lifetime of our system is extremely short, the emission-absorption processes are, as we show, highly stimulated during the red shift, also at the higher energies. This helps to fulfill the thermalization criterion of several emission-absorption cycles within the lifetime. The thermalization rates in the present case are higher (e.g., 0.20 eV/ps in Fig. 2e) than observed for the same molecular concentration in our previous work[28] (0.08 eV/ps), further confirming that the larger number of photons and stimulated processes are contributing to the speedup.

A discrete step of simultaneous creation of a low energy polariton and a vibrational quantum is often quoted as the route for organic semiconductor polariton condensation[14,29,31,57,58]. The microscopic foundation of this phenomenon is the same as that of photon condensation, but the parameter regimes differ. A

discrete step is likely to be the adequate picture when the absorption and emission spectra of the molecules show distinct vibrational sub-peaks; in contrast, a smooth red shift process leading to a BE distribution is more likely for molecules whose vibrational states are not prominently visible in the spectra[52]. Our system which shows strong coupling (polaritons), and molecule spectra with no vibrational shoulders (Fig. 1d), is in the middle ground between the weak coupling photon condensation mechanism and the relaxation by a discrete step, and requires a theoretical description going beyond the approximations done in both. For a single molecule with one vibrational state coupled to a few light modes, one can theoretically describe how a process that is coherent, except for vibrational losses that provide the energy loss channel, leads to rapid red shift of emission (Supplementary Fig. 5 and Note 6). To rigorously describe our observations, one would need a model consisting of many molecules with several vibrational states coupled to a thermal reservoir, and multiple light modes at the multiphoton regime. The model should then be solved without resorting to weak coupling perturbation theory in the light-matter coupling. The current state-of-the-art theory[59–63] forms a good starting point for this kind of advanced description. Such theory could predict how the condensation threshold depends on losses, thermalization speed, and competition with lasing, which are expected to play a role since the photon numbers emitted by the condensate that we observe here are several orders of magnitude larger than the equilibrium estimate for the critical number[28].

**Effect of pulse duration**. The spatial measurements have enabled an astounding, yet indirect, way to look into the dynamics of the system. To complement the spatial observations, we have probed the dynamics directly in the time domain by altering the excitation pulse duration. A 50 fs excitation pulse results in the double-threshold behavior with a distinct regime for lasing, an intermediate regime showing an incomplete stimulated thermalization process, and condensation, as explained above. Remarkably by using a longer pulse, we observe only the first (lasing) threshold, and the system does not undergo condensation even at higher pump fluences. The real space intensity distribution and spectrum remain almost unchanged from low to high pump fluence for a 500 fs pulse duration, Fig. 6. The intensity distributions and spectra resemble the lasing regime seen at low pump fluence with the 50 fs pulse (Fig. 2b, c), while the intermediate and condensation regimes are absent. Besides the luminescence intensity, the different threshold behavior is clearly visible in the FWHM curves of the spectral maximum (Supplementary Fig. 6 and Note 7). The FWHM is significantly decreased with both pulse durations at the first (lasing) threshold but only with the 50 fs pulse, the FWHM is decreased even further at the second (condensation) threshold. The $k$-space images and spectra for the 500 fs pulse (Supplementary Fig. 7, Movie 4) reveal that the luminescence from low to high pump fluences is spread widely in the TM mode, hence there is no 2D confinement.

So far we have compared the results for 50 and 500 fs excitation pulses at the same pump fluences so that the total amount of energy injected to the system per pulse is the same for different pulse durations. However, triggering the stimulated thermalization process might depend on the instantaneous pump intensity rather than the fluence, which means that the condensation threshold could be reached also with longer pulses if the pump fluence was increased. To test this, we have further studied the dependence on pulse duration by several intermediate measurements (Fig. 7) that show condensation with 100 and 250 fs pulses, but not with longer pulses. The condensation threshold for the 100 fs pulse is equal to that of the 50 fs pulse (3.5 mJ cm$^{-2}$),

whereas for the 250 fs pulse it is higher (4.5 mJ cm$^{-2}$). The resulting line spectra at the condensation threshold for the 100 and 250 fs pulse durations (blue and red solid lines in Fig. 7a) are similar to that measured for the 50 fs pulse presented in Fig. 2a: macroscopic population at the band edge followed by a linear distribution at the higher energies. Fit of the tail to the MB distribution gives $316 \pm 2$ and $331 \pm 4$ K for the 100 and 250 fs pulse, respectively (for more information see Section "Methods: Fits to the MB distribution"). With a longer pulse (350 fs), thermalization in the time-integrated signal remains incomplete (too much population in the higher energy states). With the longest excitation pulses (>350 fs), we observed no signs of thermalization or condensation even at the highest pump fluences that we could measure until damaging the samples. For all pulse lengths, the thermalization process competes over the same gain with the lasing. It seems that for a long excitation pulse the instantaneous population inversion does not reach a high enough value for the thermalization process to take over the lasing which is already triggered at the first threshold (see also Fig. 6).

The observations for different pulse durations are summarized by Fig. 7c, which shows the thresholds for lasing and condensation, as well as the start of the incomplete stimulated thermalization regime, as a function of pulse duration. The dependence of the condensation and incomplete stimulated thermalization thresholds on the pulse length is stronger than in the lasing case where the threshold is quite monotonous as a function of the pulse duration. Sensitivity to the excitation pulse duration highlights the ultrafast nature of plasmonic systems and endorses the sub-picosecond dynamics of the thermalization process.

Interestingly the critical pulse duration for observing condensation is similar to or smaller than the time (~250–350 fs) in which the polaritons propagate from the edges to the center in the $100 \times 100\,\mu m^2$ array (see Section "Methods: Samples for discussion on the group velocity"). Besides the critical pulse duration, observing the condensation requires that the thermalization time matches the propagation time so that the polaritons have redshifted to the band-edge energy while still having large population density. This is achieved by an optimal balance between the dye concentration and pump fluence (thermalization speed), lattice size (distance that the excitations need to propagate from the array edges to the center), and lattice period (band-edge energy). The condensation is also sensitive to the incident angle of the pump. To obtain a clear tail with MB distribution in the time-integrated luminescence signal, the pump pulse needs to come at normal incidence, and even a slight misalignment changes both the real space intensity pattern and the spectral distribution. Nonzero incidence angle results in a time difference of the excitation at the two edges of the array, and can cause an asymmetry in the populations of counter-propagating polariton modes.

## Discussion
Fundamental questions on the dynamics of BEC in driven-dissipative systems are still largely open, despite years of research[1,2,15,16]. What is the nature of the energy relaxation and thermalization processes, how does the condensate form, and what are its quantum statistical and long-range coherence properties? These questions have been addressed for weakly coupled BECs, but become challenging to answer for strongly coupled room temperature condensates as higher energy scales imply faster dynamics. Here, we have shown that plasmonic lattices offer an impressive level of access and control to the sub-picosecond dynamics of condensate formation via propagation of excitations and the finite system size.

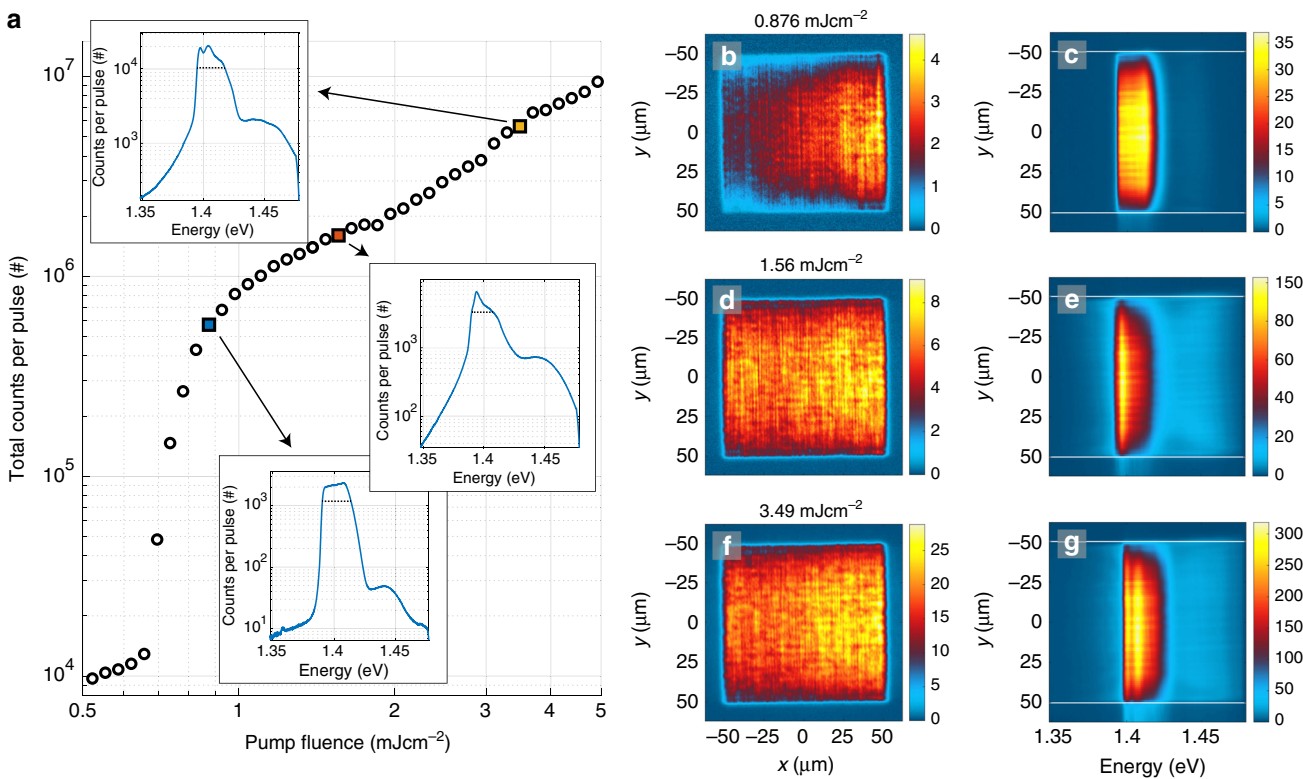

**Fig. 6 Pump fluence dependence, real space images and spectra for 500 fs pump pulse duration. a** Pump fluence dependence of the total luminescence intensity. Insets: Line spectra obtained by integrating over the real space spectra in the y-direction (between the white lines). **b–g** Left column: Real space images of the plasmonic lattice. Right column: Spectral information of the luminescence as a function of position in the y-axis. The pump fluence is **b**, **c** 0.88 mJ cm$^{-2}$, **d**, **e** 1.6 mJ cm$^{-2}$, **f**, **g** 3.5 mJ cm$^{-2}$. FWHM of the spectral peaks is marked in the insets. With increasing pump fluence, we obtain a FWHM of 23, 18, and 23 meV. The results for the full range of pump fluences are presented in Supplementary Movie 3.

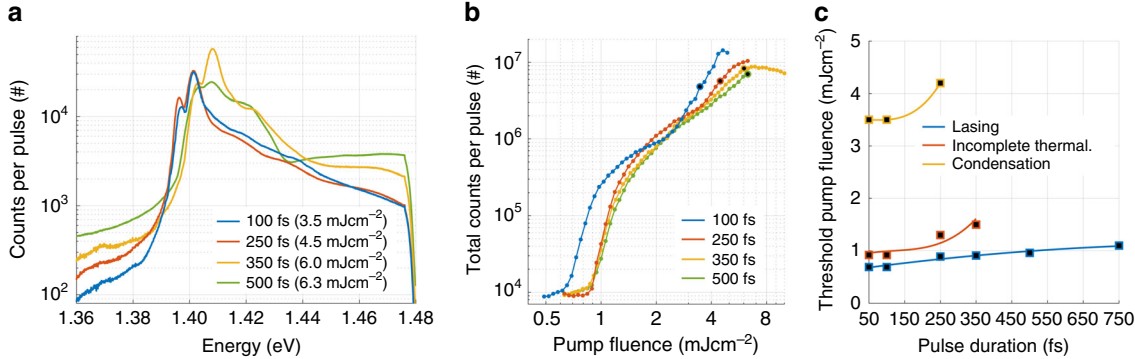

**Fig. 7 Line spectra and threshold pump fluences for different pulse durations. a** Line spectra for different pulse durations. Condensation occurs for 100 and 250 fs pulses, but not for longer ones. For the 100 and 250 fs pulses, the spectra are shown at the condensation threshold (3.5 and 4.5 mJ cm$^{-2}$, respectively). The threshold in case of the 100 fs pump pulse is the same as for the 50 fs pulse. We have defined the condensation threshold such that a narrow peak is observed at the band edge together with a linear distribution at high energies in the time-integrated signal. For a 350 fs pulse no condensation is observed, but we show a spectrum for which the tail of the distribution is closest to linear (at 6.0 mJ cm$^{-2}$). For a 500 fs pulse the spectrum is shown at the highest measured pump fluence (6.3 mJ cm$^{-2}$). **b** Total luminescence intensity as a function of pump fluence. Increasing the pump pulse duration increases slightly the first (lasing) threshold, whereas the second (condensation) threshold is visible only for the 100 and 250 fs pulses. For the 350 fs pulse, we increased the pump fluence up to 10 mJ cm$^{-2}$, after which the nanoparticle array starts to damage, to confirm that no condensation threshold occurs even at higher fluences. Rather a saturation and degradation of the luminescence signal can be observed. The highlighted data points correspond to the line spectra in **a**. **c** Threshold pump fluence for lasing and condensation, as well as the beginning of the incomplete stimulated thermalization regime, as a function of pulse duration. In the lasing regime, there is one spectral intensity maximum corresponding to the band edge (this is the lasing peak), and another one (smaller in intensity) at higher energy. First, for increasing pump fluence, the lasing peak grows and the higher energy peak diminishes. At a certain pump fluence, however, the higher energy peak starts growing. We determine this pump fluence value from the line spectra and define it as the threshold for the incomplete thermalization regime. At this pump fluence, the interference patterns (such as in Fig. 2e) become visible in the real space spectra. The lines are guides to the eye.

We experimentally demonstrated that a bosonic condensate with a clear-cut thermal excited state population can form in a timescale of a few hundreds of femtoseconds. We propose that this extraordinary speed of thermalization is possible because the process is partially coherent due to strong light–matter coupling and stimulated emission. Strong light–matter coupling at the weak excitation limit was indicated by our reflection measurements. By varying the lattice size, we revealed the stimulated nature of the thermalization process. While strong light–matter coupling at the multiphoton regime is described by the Dicke model[64], to explain both the red shift and the thermal distribution we observed, one would need to involve vibrational degrees of freedom that are (strongly) coupled to the electronic ones as well as to a thermal bath. Work toward surmounting such theoretical challenges has already begun since systems where light and molecular electronic and vibrational states are strongly coupled have promise for lasing and condensation, energy transfer, and even modification of chemical reactions[65,66]. We have shown here that plasmonic lattices offer a powerful platform for studies of such ultrafast light–matter interaction phenomena. The shape, size, and material of the nanoparticles can be accurately tuned, as well as the lattice geometry, composition of the unit cell, and the lattice size. This provides a large, controlled parameter space vital for testing and (dis)qualifying theoretical predictions. In particular, the dynamics can be accessed in complementary ways: through conventional time-domain techniques as well as indirectly via the propagation of excitations in the lattice.

## Methods

**Samples**. The gold nanoparticle arrays are fabricated with electron beam lithography on glass substrates where 1 nm of titanium is used as an adhesion layer (see SEM image in Supplementary Fig. 8). The nominal dimensions of the plasmonic lattice, optimized for the condensation experiment, are the following: a nanoparticle diameter of 100 nm and height of 50 nm, the period in $y$- and $x$-direction of $p_y = 570$ and $p_x = 620$ nm, respectively, and a lattice size of $100 \times 100$ μm². In reference measurements, the period $p_y$ is varied between 520 and 590 nm and the lattice size between $40 \times 40$ and $200 \times 200$ μm².

Asymmetric periodicity separates the diffracted orders in the energy spectrum for the two orthogonal polarizations ($\mathbf{e}_x$ and $\mathbf{e}_y$), and the SLR dispersions are correspondingly separated, which simplifies the measured spectra. For $x$-polarized nanoparticles (as in our experiments), the TE and TM modes correspond to combinations of ($\mathbf{e}_x, k_y$) and ($\mathbf{e}_x, k_x$), respectively. Under pumping, which SLR mode is mainly excited is determined by the pump polarization as the molecules are excited more efficiently via the single particle resonance at the plasmonic hotspots of each nanoparticle[41]. In the experiment with different periods, $p_x$ was kept always 50 nm larger than $p_y$. In the experiment where lattice size was varied, however, the lattice period in $x$ and $y$ directions was the same ($p_x = p_y = 570$ nm). We found that asymmetric periodicity does not play a crucial role in forming the condensate but just simplifies the data analysis of the experimental results.

Group velocity for the TE mode is obtained close to the $\Gamma$-point, in the linear part of the dispersion, for samples both without and with the dye molecules. The group velocity is $0.65c$ for the uncoupled TE mode (Fig. 1c) and $0.48c$ for the strongly coupled polariton mode (Fig. 1d; $c$ is the speed of light). We use the group velocity to convert propagation distance to time. In the experiments we see that the strongly coupled dispersion starts to resemble the uncoupled one at high pump fluences due to the saturation effects, as explained in the manuscript. We cannot exactly specify what the group velocity is at a certain pump fluence, therefore we present the time conversions with a group velocity range from $0.48c$ to $0.65c$. This means that the propagation of 50 μm distance along the array takes 250–350 fs.

The dye molecule solution is index-matched to the glass substrate ($n = 1.52$), the solution being a mixture of 1:2 DMSO:Benzyl Alcohol. The solution is sealed inside a Press-to-Seal silicone isolator chamber (Sigma-Aldrich) between the glass substrate and superstrate. The dye solution has a thickness of ~1 mm, which is very large compared to the extent of the SLR electric fields[27,39,41]. Given by the excess of the dye molecules and the natural circulation of the fluid, there are always fresh dye molecules available for consecutive measurements of the sample when scanning the pump fluence, which makes the sample extremely robust and long-lasting. IR-792 perchlorate ($C_{42}H_{49}ClN_2O_4S$) was chosen as the dye molecule because it dissolves to the used solvent in very high concentrations, in contrast to many other dye molecules, e.g., IR-140 that has also been used by us[27,39] and others[24,26,41] as a gain medium in plasmonic nanoparticle array lasers. We have collected the information on tested dye molecules and clarify the reasoning behind the molecule choice in Supplementary Table 1.

**Transmission, reflection, and luminescence measurement setup**. A schematic of our experimental setup is depicted in Supplementary Fig. 9. The same setup is used for transmission, reflection and luminescence measurements with minor modifications. The spectrometer resolves the wavelength spectrum of light that goes through the entrance slit, and each pixel column in the 2D CCD camera corresponds to a free space wavelength, $\lambda_0$, and each pixel row to a $y$-position at the slit. The $y$-position further corresponds to either an angle ($k$-space) or the $y$-position at the sample (real space). The photon energy is $E = hc/\lambda_0$ and in the case of angle-resolved spectra (dispersions) the in-plane wave vector $k_{x,y} = k_0 \sin(\theta_{x,y}) = 2\pi/\lambda_0 \sin(\theta_{x,y})$, where $h$ is the Planck constant and $c$ the speed of light in free space. Next, the three different experiment types are explained, starting with the luminescence measurement where the sample is optically excited with an external pump laser. The excitation pulse (or pump pulse) is generated by Coherent Astrella ultrafast Ti:Sapphire amplifier. The pulse has a central wavelength of 800 nm, and at the laser output, a duration of <35 fs with a bandwidth of 30 nm. The pump pulse is guided through a beam splitter and mirrors, and finally to the mirror M1 (see Supplementary Fig. 9), which directs the pump pulse to the excitation path of the setup. We have a band-pass filter in the excitation path that is used in combination with a long-pass filter in the detection path to filter out the pump pulse in the measured luminescence spectra. The pump pulse is linearly polarized, and to filter only the horizontal component we have placed a linear polarizer after the band-pass filter. The pump fluence is controlled with a metal-coated continuously variable neutral-density filter wheel (ND wheel). The pump pulse is spatially cropped with an adjustable iris and the iris is imaged onto the sample with a help of lens L1 and the microscope objective. The inverted design enables exciting the sample at normal incidence, which is crucial for simultaneous excitation of the dye molecules over the sample. Excitation at normal incidence also prevents any asymmetry in the spatial excitation of the molecules around the nanoparticles with respect to lattice plane. The inverted pumping scheme and accurate optical alignment were essential for repeatable and precise condensate formation.

In the detection path, we have the long-pass filter and optionally a linear polarizer. An iris or pinhole acts as a spatial filter at the 1st image plane to restrict the imaged area at the sample. The 1st image plane is relayed onto the real-space camera (1st Cam.). In the $k$-space measurements, the back-focal plane of the objective (Fourier plane; containing the angular information of the collected light) is relayed onto the 2D $k$-space camera (2nd Cam.) as well as onto the spectrometer slit, with the tube lens and a $k$-space lens. For the real space measurements, the beam-splitter before the $k$-space lens is replaced with an additional real-space lens to produce the 2nd image plane of the sample to 2nd Cam. and onto the spectrometer slit. The spectrometer slit selects a vertical slice either of the 2D $k$-space image or the real space image. In the luminescence measurements, we use a slit width of 500 μm. In the $k$-space, it corresponds to ±1.3° around $\theta_x = 0$, or to ±0.16 μm⁻¹ around $k_x = 0$ at $E = 1.4$ eV. Respectively in the real space, the slit opening of 500 μm corresponds to 27 μm slice at the sample.

The dispersion of optical modes in the bare plasmonic lattice can be measured in transmission mode of the setup, where the sample is illuminated with a focused and diffused white light from a halogen lamp. The lattice modes are visible as transmission minima (extinction maxima) in the angle-resolved spectrum. When a thick layer of high-concentration dye molecule solution is added in the chamber on top of the nanoparticle array, the transmission measurement is not applicable due to a complete absorption by the molecules. To access the dispersion of the lattice modes in this case, we use the setup in reflection mode by utilizing the same inverted design as in the luminescence measurement. The halogen lamp is inserted before the iris in the excitation path, that is imaged onto the sample, and the dispersion of the lattice modes is revealed by reflection (scattering) maxima in the collected angle-resolved spectrum.

The luminescence measurement as a function of pump fluence is automated with LabView. Predefined fluence steps are measured such that for each step: (1) the calibrated ND filter wheel is set to a correct position, (2) the shutter is opened, (3) the image is acquired with spectrometer and optionally with 1st and 2nd Cam., and (4) the shutter is closed. Integration time of the spectrometer is automatically adjusted during the measurement to avoid saturation at highly non-linear threshold regimes. The pump pulse duration is measured with a commercial autocorrelator (APE pulseCheck 50). In the pulse duration measurement, the pump pulse goes through the same optics as in the actual experiments. The pulse duration is changed by adjusting the stretcher-compressor of the external pump laser.

**Fits to the MB distribution**. We fit the thermalized tail in the measured population distributions to MB distribution (Fig. 2, Supplementary Fig. 6). The fit function is given by

$$f_{MB}(E) = \frac{d(E)}{e^{(E-\mu)/(k_B T)}}, \tag{1}$$

where $d(E)$ is the degeneracy of the modes as a function of energy $E$, $\mu$ is the chemical potential, $k_B$ is the Boltzmann constant, and $T$ is temperature. The fit was done for the part of the distribution that is linear in logarithmic scale, at pump fluence at/above the threshold. We call this part of the distribution (between energies 1.41 and 1.47 eV) the "tail". Fitting was performed with a nonlinear least squares method.

The degeneracy $d(E)$ was approximated by the density of states for light traveling in a 2D plane. The light dispersion in the $xy$ plane forms a conical surface $(E = \hbar c/n \sqrt{k_x^2 + k_y^2})$. The dispersions of the SLR modes are well approximated by this everywhere except the very near vicinity of the $\mathbf{k} = 0$ point[67], and our fitted range starts from a finite $\mathbf{k}$ where the approximation is valid. This dispersion results in a linearly increasing but nearly constant density of states for the fitted energy range of ~60 meV ($d(E) = 1. . . 1.05$).

The fit gives a temperature of $313 \pm 2$ K for a chosen pump fluence of $P = 3.5$ mJ cm$^{-2}$, error limits representing the 95% confidence bounds for the fit. This fit is presented in manuscript Fig. 2a (top inset) and Supplementary Fig. 6a. The fitted pump fluence is chosen such that the fluence is the lowest showing a linear slope in the time-integrated population distribution (in logarithmic scale). The chosen fluence also corresponds to the narrowest FWHM of the highest condensate peak (see Supplementary Fig. 6c). The high-energy tail remains linear over pump fluences between ~3.5 and 4 mJ cm$^{-2}$, with a slightly changing slope. For two higher pump fluences of 3.7 and 3.9 mJ cm$^{-2}$, the fits to the the Maxwell-Boltzmann distribution give temperatures of $282 \pm 2$ and $250 \pm 2$ K, respectively. Goodness of fit is described by the square root of the variance of the residuals (RMSE) and the R-square value. The values of (RMSE, R-square) for the three pump levels 3.5, 3.7, and 3.9 mJ cm$^{-2}$ are (107, 0.996), (98, 0.998), and (146, 0.998), respectively. For pump fluences above ~4 mJcm$^{-2}$, the linear slope is distorted and the condensate degrades. This is evident also as a decrease of the spatial coherence (Fig. 4a) and an increase of the FWHM of the spectral maximum (Supplementary Fig. 6c).

For the longer pulses, 100 and 250 fs, the fit gives $316 \pm 2$ and $331 \pm 4$ K at the condensation threshold, 3.5 and 4.5 mJ cm$^{-2}$, respectively. The values of (RMSE, R-square) for the fits are (119, 0.997) and (219, 0.985), respectively. The fits are still quite good for these pulse durations, whereas for 350 fs and longer pulses no thermal MB distribution was observed.

**Estimation of photon number in the condensate**. The photon number in the condensate is estimated from the measured luminescence intensity. A strongly attenuated beam from the external pump laser (800 nm, 1 kHz) is directed to the spectrometer slit, and the total counts given by the spectrometer CCD camera (Princeton Instruments PIXIS 400F) is compared to the average power measured with a power meter (Ophir Vega). The measured average power of 167 nW corresponds to $6.7 \times 10^8$ photons/pulse whereas the total counts in the CDD camera are $8.4 \times 10^6$, leading to a conversion factor of ~80 photons/count. In the condensation regime, the total counts per pulse is about $3 \times 10^6$ (manuscript Fig. 2a). The collection optics including the beam splitters reduce the signal roughly by a factor of 2.5, and as the slit width of 500 μm corresponds to 27 μm at the sample, we collect luminescence from an area that is about 1/4 of the 100 μm wide nanoparticle array. Finally, the sample is assumed to radiate equally to both sides, so the actual photon number per emitted condensate pulse becomes: $n_{\mathrm{ph}} \approx 2.5 \times 4 \times 2 \times 80 \times 3 \times 10^6 = 4.8 \times 10^9$.

**Spatial coherence measurements**. Spatial coherence of the sample luminescence is measured with a Michelson interferometer. The real space image is split into two arms and the image in one of the arms is inverted in vertical direction with a hollow roof retro-reflector. Then the two images are combined again with a beam splitter and overlapped at the camera pixel array, simultaneously. With this design the spatial coherence (first-order correlation $g^{(1)}$) can be measured separately along both $x$- and $y$-axis of the plasmonic lattice. The retro-reflector always inverts the image vertically, with respect to $y = 0$ in laboratory reference frame, but the sample and the pump polarization can be rotated 90° to measure $g^{(1)}(-\mathbf{x}, \mathbf{x})$ instead of $g^{(1)}(-\mathbf{y}, \mathbf{y})$ in the lattice coordinates. The first-order correlation function describing the degree of spatial coherence is given by

$$g^{(1)}(-\mathbf{y}, \mathbf{y}) = \frac{\langle E^*(-\mathbf{y})E(\mathbf{y})\rangle}{\sqrt{\langle E(-\mathbf{y})^2\rangle \langle E(\mathbf{y})^2\rangle}}, \qquad (2)$$

where $E(\mathbf{y})$ is the electric field at point $\mathbf{y}$. The first-order correlation function relates to the interference fringe contrast $C$ as follows:

$$C(\mathbf{y}, -\mathbf{y}) = \frac{2\sqrt{I(\mathbf{y})I(-\mathbf{y})}}{I(\mathbf{y}) + I(-\mathbf{y})} g^{(1)}(\mathbf{y}, -\mathbf{y}), \qquad (3)$$

where $I(\mathbf{y})$ is the luminescence intensity at point $\mathbf{y}$ of the lattice. The fringe contrast in the interfered images is extracted with a Fourier analysis as explained in Supplementary Note 4.

## Data availability
The data that support the findings of this study are available in zenodo.org with the identifier DOI: 10.5281/zenodo.3648650[68].

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

## Acknowledgements

We thank Janne Askola for his help in intensity calibration of the spectrometer. We thank Jonathan Keeling and Kristín Arnardóttir for useful discussions. This work was supported by the Academy of Finland under project numbers 303351, 307419, 327293, and 318987 (QuantERA project RouTe), 322002, 320166, and 318937 (PROFI), by Center for Quantum Engineering (CQE) at Aalto University, and by the European Research Council (ERC-2013-AdG-340748-CODE). This article is based on work from COST Action MP1403 Nanoscale Quantum Optics, supported by COST (European Cooperation in Science and Technology). Part of the research was performed at the Micronova Nanofabrication Center, supported by Aalto University. The Triton cluster at Aalto University (Science-IT) was used for computations. A.J.M. acknowledges financial support by the Jenny and Antti Wihuri Foundation. K.S.D. acknowledges financial support by a Marie Skłodowska-Curie Action (H2020-MSCA-IF-2016, project ID 745115).

## Author contributions

P.T. initiated and supervised the research. A.I.V. fabricated the samples. A.I.V., A.J.M., and K.S.D. conducted the experiments. A.J.M., A.I.V., P.T., and T.K.H. did the data analysis. A.I.V. performed the rate-equation, A.J.M. the quantum model, and M.N. the T-matrix calculations. A.I.V., A.J.M., and P.T. wrote the paper with all authors.

## Competing interests

The authors declare no competing interests.
