## [Peer Review File · Nature Communications]

Reviewers' comments:

Reviewer #1 (Remarks to the Author):

Dear editor and authors,

the article 'sub-ps thermalization dynamics in lattice plasmons' by Väkeväinen and co-authors reports on studies of a plasmonic system coupled to dye molecules, with a focus on the thermalization dynamics. I think the results and the experimental technique that allows to investigate extremely fast thermalization and condensation dynamics are interesting, and the results obtained in the thermal regime (a Boltzmann behaviour for $2 k_B T$) is quite impressive.

I also think that the central questions the paper tries to address are interesting, such as the mechanism behind the thermalisation processes and the formation of the condensate. However, the discussion is not very clear for the most part, and surely not suitable for the broad readership of nature communications. Additionally, many interesting data that help understanding the details are only found in the supplementary, so some restructuring of the manuscript might be in order.

Let me address a few of the points that should be explained in more detail:

i) The dye molecules have an upper state lifetime of a few nanosecond, which means that spontaneous emission can be neglected completely, and stimulated emission will only become relevant for highly populated modes. Correspondingly, the thermalization must be at least partly driven by the polariton-polariton interaction, especially in the thermal cloud where the population density is low. As the very fast dynamics of the thermalisation in itself is very interesting and I think it would help if the physical mechanisms are discussed with more detail. For example, what are estimated time scales that affect the thermalisation time? Or are the dye molecules in this system only relevant to shape the dispersion relation, such that propagation speed, thermalization time, loss rates etc. are in the correct regime?

In the same line of reasoning: The TE-mode is not confined, i.e. serves as a loss channel. If thermalization proceeds via the dye molecules, which do not discriminate between TE and TM, then why does this loss channel still allow for a condensate? This goes along with the photon number: The condensate contains roughly 10^9 photons, while the pump pulse delivers (at the condensation threshold) roughly 10^{12} photons to the sample. First, can the authors give estimates of the expected threshold, assuming this would be an equilibrium system? Second, can the ratio of pump to condensate photons be used to extract some information on the losses?

ii) The fact that for longer pump pulses no condensate is achieved does not seem surprising to me: The existing loss mechanisms lead to a reduction of the number of polaritons in the system, and for pulses longer than the typical loss rate the average polariton number will be below the critical polariton number. Thus, by increasing the pump fluence one should see a condensate again. I assume experimentally this is not feasible? However, by observing the fluence required for condensation for different pulse length (as is done in Fig. S10 for one fluence) one should be able to extract information on the loss rates in the system. At the same time, one should check if the number of polaritons at the condensation threshold is a constant, which I would expect for a close-to-equilibrium system, or if there is some dependence on the external parameters.

iii) The observed thresholds seem a bit ambiguous, as they are only valid for this exact system. As it is already shown that changing the pump pulse duration affect the observations (lasing or condensation) greatly, it would also help if the dependence of the observed double threshold behaviour to other experimental parameters such as pump geometry or lattice size is discussed.

iv) I am not convinced if distinguishing the stimulated thermalization regime so strongly is justified. From the observed energy distributions, this regime seems to be a regime of imperfect

thermalisation prior to condensation? Or more concretely, what (experimental) criterion is used to call a spectrum thermalized? One could e.g. look at the deviation from a Boltzmann distribution and constrain this somehow to have a quantitative measure.

v) In the previous experiments of the research group a single-mode condensate was achieved. Thus, more discussion on the multimode behaviour in the condensed regime is also necessary. There is some information about possible explanations of this in the supporting file. However, the discussion will be much more clear if the simulated mode structure of the lattice is plotted beside the observed spectra. Also, are there any other possible explanations for the observed multimode behaviour? Why is it not simply multi-mode lasing? The definition of a multimode condensate is anyway to be taken with a grain of salt, as most people take macroscopic population of the ground state as "the condensate", thus multimode condensates in itself are not governed by this Definition, so some discussion is required.

vi) It is interesting that by tuning the pump pulse duration and the lattice size, a spectral distribution that very closely matches Maxwell-Boltzmann distribution can be attained. I think the fact that this equilibrium distribution is achieved outside a steady state deserves more discussion, apart from the short discussing on page 14 and in the supplementary S5. While in a thermal equilibrium system the condensate is always achieved, here it seems more like an interplay between losses and gain, similar to the experiments on classical condensation by e.g. Baruch Fischer, and thus I would hope for a longer discussion on the physics. Additionally, some comments on which thermodynamic quantities are expected to follow predictions for thermal equilibrium in such a case apart from the spectrum would help.

vii) The conclusions contain claims that are not supported I believe: The authors make several claims that have already been addressed for photon condensates, mostly by the Weitz group:
-"What is the nature of the energy relaxation, how does the condensate form" -> Ref. 32
- "What are the quantum statistical properties" -> Ref. 21 and Nature communications 7, 11340 (2016)
- "long-range coherence properties" -> PRL 116, 033604 (2016), Nature communications 8 (1), 158 (2017)

Thus, these claims are not really supported in general. Of course they are still true for the polariton-dye system, but not for photonic BECs per se.

viii) The abstract contains the claim "Our results give us direct access to spatiotemporal control and monitoring of thermalization processes...". While I agree with the control, which is here obtained by changing the pulse length, the "monitoring" part is to my opinion not shown in the paper. While for the propagating system as in previous work of the Törmä group a direct matching between position and time can be made, here a temporal monitoring would require a very sophisticated solution. There are streak cameras with 100 fs resolution available, but even this resolution obviously does not suffice.

Thus, in general I think the results of the paper are very interesting and deserve publication. However, in the current version of the manuscript the quite general claims made are not supported by the data, or not explained such that the reader could easily connect the data to the claims. Correspondingly, for publication in nature communications major revisions are needed to my belief. Most importantly, a clear explanation of the thermalization process, see my point (i), is required, as only then the reader can understand the interplay of the different mechanisms involved.

Correspondingly, the paper can only be accepted after major revisions, or should be sent to a more specialized Journal after revisions are made.

Reviewer #2 (Remarks to the Author):

The manuscript "Sub-picosecond thermalization dynamics in condensation of strongly coupled lattice plasmons" by Väkeväinen et al. demonstrates ultrafast thermalization dynamics of polariton lasing and condensation in a plasmonic nanoparticle array with strong light-matter coupling. In particular, the authors find three distinct regimes of light emission separated by two thresholds, with the first threshold corresponding to a one-dimensional lasing regime and the second threshold, which is only observed for short enough pulse durations, assigned to a two-dimensional multimode condensation regime including thermalization. These results are very interesting and present a novel phenomenon that to my knowledge has not been observed in these or similar systems before. The work thoroughly explores the different regimes and provides rich insight into the properties and dynamics of the emitted light. I thus wholeheartedly recommend publication in Nature Communications, provided the following comments are addressed:

- 1) The authors mention that they achieve a signal that is 5 orders of magnitude larger than in their previous Nature Physics paper (ref. 28), which enables the observation of a thermal tail after condensation. However, it is not clear to me how or why this is achieved and what is the crucial difference with respect to the previous setup.
- 2) It is also not fully explained why the authors refer to the regime reached after the first threshold as lasing and after the second as condensation - this seems to rely on the observation of a thermalized tail after the second threshold, but the reasoning behind this should be clarified.
- 3) The Michelson interferometer measurements seem very similar to those recently reported in reference 30, which also showed the stripes observed in the "1D lasing" regime identified here. It would be helpful to a reader not familiar with the literature to mention this. Also, could further insight be provided into why these 1D stripes show up and full 2D coherence is not reached in this regime?
- 4) The continuous redshift of the emission during the propagation implies a somewhat continuous thermalization process, which seems slightly at odds with the picture of (resonant) vibration-driven decay between states that is often invoked in organic molecular polariton condensation. For example, this is identified in a very recent article (Ramezani et al, Nano Letters 2019, <https://doi.org/10.1021/acs.nanolett.9b03139>, which should probably be cited) as the main pathway for polariton decay, also on ultrafast timescales. Can it be understood where this seeming difference between decay driven by discrete vibrational modes and continuous redshifting comes from?
- 5) After the second threshold, the "condensate" peak has a multi-peak substructure in energy (see, e.g., Fig 3j, lowest panel). This is explained in the supplemental (section 6) as due to an approximate degeneracy between different modes that is lifted for finite-sized nanoparticles. However, it did not become clear to me what these different modes correspond to, maybe this could be clarified?

Reviewer #3 (Remarks to the Author):

The manuscript reports about an experimental observation of the sub-picosecond thermalization dynamics in the condensation formation of strongly coupled lattice plasmons. The high quality and the level of detail of the obtained results as well as their concise presentation justify that the manuscript is published at Nature Communications. However, prior to publication, the authors should respond to the following questions and comments :

1) Is it possible to estimate the magnitude of the photon-photon interaction and to determine at least qualitatively how it depends upon the concentration of the dye molecules?

2) Which criteria justify to choose the particular organic dye molecule IR-792? Which other dye solutions could, in principle, be used?

3) On the one hand, at the beginning the authors mention the topic of Bose-Einstein condensation of photons, but then on page 7 they argue to have measured only a Maxwell-Boltzmann distribution.

Thus, the general question arises whether the observed condensation of strongly coupled lattice plasmons is a quantum or a classical effect. This question reminds one to a similar situation, which occurs for the condensation of magnons, where the original experimental publication

Nature 443, 430 (2006)

claims to have seen a Bose-Einstein condensation, but later on this was more precisely identified to be just a Rayleigh-Jeans condensation in

Phys. Rev. Lett. 115, 157203 (2015)

4) Another question concerns symmetry arguments. From Fig. S1 the nanoparticle array is symmetric in x- and y-direction. On the other hand

a) it is argued on page 9 that lasing exhibits confined luminescence along k_y but spreads along k_x

b) Fig. 4 reveals that spatial coherence is asymmetric in x- and y-direction.

What is the physical reason for that asymmetry?

5) The authors should mention in the text the order of magnitude of the coherence length and how to explain at least the order of magnitude.

6) On page 11 the authors mention their rate equations in Section S8 and emphasize that they are not sufficient for describing the condensate or lasing observations. This raises the question how the rate equations would have to be extended to cover at least qualitatively all the experimentally observed features.

7) The authors mention on page 13 that the Kirton-Keeling dynamics in Section S9 is not sufficient to describe the observed quantum dynamics. And they attribute this to the high coherence of the thermalization

due to the observed stimulated processes and due to strong light-matter coupling. Therefore I suggest that the authors apply an extension of the Kirton-Keeling dynamics to their system which takes into account

self-consistently not only the dissipative but also the coherent dynamics :

New J. Phys. 20, 055014 (2018)

8) At the end of their manuscript the authors discuss the effect of pulse duration. Here I suggest

to make the manuscript more concise by producing and discussing a phase diagram, where the pump fluence for the occurrence of the first and second threshold are plotted versus the pump duration. Such a phase diagram would summarize within one figure various experimental results, which are currently distributed in different figures.

Response to the Reviewer reports

Sub-picosecond thermalization dynamics in condensation of strongly coupled lattice plasmons

Aaro I. Väkeväinen, Antti J. Moilanen, Marek Nečada,
Tommi K. Hakala, Konstantinos S. Daskalakis, Päivi Törmä

Reviewer: 1

Dear editor and authors, the article 'sub-ps thermalization dynamics in lattice plasmons' by Väkeväinen and co-authors reports on studies of a plasmonic system coupled to dye molecules, with a focus on the thermalization dynamics. I think the results and the experimental technique that allows to investigate extremely fast thermalization and condensation dynamics are interesting, and the results obtained in the thermal regime (a Boltzmann behaviour for $2 k_B T$) is quite impressive.

I also think that the central questions the paper tries to address are interesting, such as the mechanism behind the thermalisation processes and the formation of the condensate. However, the discussion is not very clear for the most part, and surely not suitable for the broad readership of nature communications. Additionally, many interesting data that help understanding the details are only found in the supplementary, so some restructuring of the manuscript might be in order.

We thank the reviewer for stating that the central questions of our paper are interesting and the results impressive. We also thank for the careful reading of our manuscript and the deep questions presented by the reviewer. We have revised the manuscript to address these points and to make our manuscript suitable for the broad readership of Nature Communications. Please see below our detailed answers.

Let me address a few of the points that should be explained in more detail:

i) The dye molecules have an upper state lifetime of a few nanosecond, which means that spontaneous emission can be neglected completely, and stimulated emission will only become relevant for highly populated modes. Correspondingly, the thermalization must be at least partly driven by the polariton-polariton interaction, especially in the thermal cloud where the population density is low. As the very fast dynamics of the thermalisation in itself is very interesting and I think it would help if the physical mechanisms are discussed with more detail. For example, what are estimated time scales that affect the thermalisation time? Or are the dye molecules in this system only relevant to shape the dispersion relation, such that propagation speed, thermalization time, loss rates etc. are in the correct regime?

We thank the reviewer for bringing up all the questions in i), they have made us to write whole new paragraphs in the revised manuscript (three last paragraphs of the "Stimulated nature of the thermalization" section) where these central issues are now explained. In our system, at the incomplete thermalization and BEC regimes of pumping, stimulated emission is actually relevant also for the higher energy modes (the "thermal cloud"). This is shown by the results in the section "Stimulated nature of the thermalization", in particular Figure 5. Therefore we do not necessarily need polariton-polariton interactions to drive the thermalization process. In the three new paragraphs, we now discuss in more detail the physical mechanism of the thermalization in our system, as requested by the reviewer; please read those paragraphs for answers to the above questions. Here just briefly: The speed of the thermalization process depends on the number of molecules, strength of the light-matter coupling, and the number of photons since stimulated processes are important. The dye molecules thus affect the thermalization time, but they do not change the propagation speed in a major way (from the dispersions, we extract a change of the group velocity from 0.65c

for bare samples to around 0.48c for those with 80 mM molecule concentration). We thank the reviewer pointing this out, since we had used the bare sample group velocity when relating distance to time in the original manuscript. We note that in the experiments the strongly coupled dispersion starts to resemble the uncoupled one at high pump fluences due to the saturation effects, as explained in the manuscript. Therefore, we cannot exactly specify what the group velocity is at a certain pump fluence. We now present the time conversions with a group velocity range from 0.48c to 0.65c. This means that the propagation of $50\mu\text{m}$ distance along the array takes between 250–350 fs. We have updated the time conversions in the manuscript and added the explanation to the Methods subsection "Samples" (Supplementary Section S1 in the previous version). The resulting changes are not significant. The molecules also alter the loss rates of the modes by about a factor of two close to the Γ -point where the condensation occurs, which can be seen by comparing the linewidths of the modes in Figs. 1c (no molecules) and 1d (molecules). Closer to the molecule absorption maximum (the white horizontal line) there is significant broadening, because the molecule component of the polariton compared to the SLR one is larger there.

Polariton-polariton interaction can originate from Coulomb interaction of the excitonic part (the molecular part) of the polariton, and such repulsive interaction would show up as a blueshift of the condensate energy with increasing pump fluence (increasing number of polaritons). Also polariton - reservoir exciton interactions can contribute to the blueshift. On the other hand, an effective interaction can arise also from saturation effects, in other words, when the system has a higher number of excitations, the collective light-matter interaction decreases; such effects can be theoretically shown to lead to an effective repulsion between polaritons (and between polaritons and reservoir excitons), which also shows up as a blueshift. Extracting the interaction constant from the observed blueshift is thus tricky, since various types of Coulomb and saturation effects contribute, moreover, the functional dependence of the blueshift on the polariton number can be complicated. However, one may make a mean-field approximation estimate for low polariton density (linear regime) (Deng et.al., Reviews of Modern Physics 82, 2010). In this case the blueshift ΔE is linearly dependent on the polariton density n and the interaction constant g , $\Delta E = gn$. Putting in our experimentally observed values of $\Delta E \sim 20$ meV and 10^{17} polaritons/ m^2 we obtain a value of $g \sim 0.2\mu\text{eV}\mu\text{m}^2$. We believe most of the effective interaction in our case is due to the saturation effects, that is, degradation of the light-matter coupling, since the whole dispersion blue-shifts. We expect the Coulombic effects are of a much smaller order of magnitude, as is typical in organic semiconductor polariton condensates. For instance, in the organic polariton condensate of Daskalakis et al. Phys. Rev. Lett. 115, 035301 (2015), the Coulomb interaction -caused polariton-polariton interaction was estimated to be $10^{-3}\mu\text{eV}\mu\text{m}^2$, in contrast to inorganic microcavity polaritons with $\sim 1\text{-}10\mu\text{eV}\mu\text{m}^2$ (e.g., Fig. 5 in Estrecho et al., Phys. Rev. B 100, 035306 (2019)). Further analysing the blueshift, taking into account all the above mentioned complexities, and understanding the nature of interactions in our system is beyond the scope of the present manuscript. An interested reader can easily calculate the $g = \Delta E/n$ from the numbers we give in the revised manuscript (Fig.2 gives the number of photons observed from the $100\mu\text{m} \times 100\mu\text{m}$ sample, which reveals n , and the blueshift can be obtained from the numbers given in the paragraph "In Fig.4,").

In the same line of reasoning: The TE-mode is not confined, i.e. serves as a loss channel. If thermalization proceeds via the dye molecules, which do not discriminate between TE and TM, then why does this loss channel still allow for a condensate? This goes along with the photon number: The condensate contains roughly 10^9 photons, while the pump pulse delivers (at the condensation threshold) roughly 10^{12} photons to the sample. First, can the authors give estimates of the expected threshold, assuming this would be an equilibrium system? Second, can the ratio of pump to condensate photons be used to extract some information on the losses?

The TE-mode is not the loss channel relevant for thermalization and condensate formation. The energy that must be lost to allow thermalization goes into the vibrational degrees of freedom of the molecules, as is now explained in the revised manuscript (the three new paragraphs).

The estimate of the critical number in an equilibrium system was given in our previous work, Hakala

et al., Nat. Phys. 14, 739 (2018) (the supplementary information there). Using the same estimate, but now with the system size (the coherence length) $L = 24\mu\text{m}$ instead of the $L = 45\mu\text{m}$ used there, we get the critical density of about $10^{11}/\text{m}^2$. Our observed condensate photon numbers, divided by the sample area, give $10^{17}/\text{m}^2$ which is orders of magnitude higher. This is understandable as photon densities higher than the equilibrium estimate are needed to compensate for losses, to stimulate the thermalization process so that it occurs fast enough, and to win over the competing process of lasing which occurs in our system at lower pump fluences. An advanced theoretical description, as we explain in the manuscript (again, the three new paragraphs), would be needed to make an estimate of the threshold which includes these factors.

It is a good question where are most of the 10^{12} photons going as the condensate contains about 10^9 . Our nanoparticle array samples are combined with a thick molecule layer, and we pump them from the top of the array, see Fig.1a): pumping occurs from the same side where we observe the luminescence, that is, we pump through the objective. The pump beam comes with such an angle (normal incidence) and energy that it does not correspond to any energy in the nanoparticle array dispersion, and thus is not strongly absorbed by the array. The pump light is coupled in to the molecules and also to some extent to the nanoparticles via the single nanoparticle resonance. The molecules have a finite cross-section, so not all of them get excited, but with increasing layer thickness, increasing amount of photons are absorbed. However, of the molecules only those that are in near vicinity ($0.5 - 1\mu\text{m}$) of the array can efficiently couple to it and thus contribute to the condensate (the stronger contribution of molecules close to the array has been demonstrated in several experimental and theoretical studies (Yang et al., ACS Nano 9, (2015), Wang et al., Nature Nanotech. 12, 889 (2017) (in particular the supplementary information), Hakala et al., Nature Commun. 8, 13687 (2017), Trivedi et al., PRA 96, 053825 (2017), Daskalakis et al., Nano Lett. 18, 4, 2658 (2018), Wang et al., Nano Letters 18, 7 (2018), Wang et al., Chemical Reviews 118, 2865 (2018), Wu et al., Advanced Optical Materials 7, 1900334 (2019), Wang et al., J. Phys. Chem. Lett. 10, 33013306 (2019), Ao et al., ACS Photonics 6, 2612 (2019)). Thus, a large part of the pump photons are lost in exciting molecules that will not emit back to the array, and we expect this to be the main cause of the difference between 10^{12} and 10^9 , although surely also losses intrinsic to the system of interest (the nanoparticle array and the nearby molecules) exist. One way to increase the ratio of absorbed pump fluence could be to have a thinner molecule layer and a mirror behind it. In fact we have fabricated samples utilizing this type of ideas and observed much lower thresholds for lasing (unpublished). However, the thick dye layer has an important practical advantage as it makes the samples persistent against photo bleaching. We have added to the manuscript an explanation (end of the section "System") stating that the pump does not directly couple to the SLR modes, and that only small fraction of the pump photons are absorbed by the "system of interest" namely the nanoparticles and the molecules in their close vicinity. We also moved the sentence: "Active region of the dye lies within a few hundred nanometers from the lattice plane..." from the Supplementary information to the manuscript, and added some of the above references.

ii) The fact that for longer pump pulses no condensate is achieved does not seem surprising to me: The existing loss mechanisms lead to a reduction of the number of polaritons in the system, and for pulses longer than the typical loss rate the average polariton number will be below the critical polariton number. Thus, by increasing the pump fluence one should see a condensate again. I assume experimentally this is not feasible? However, by observing the fluence required for condensation for different pulse length (as is done in Fig. S10 for one fluence) one should be able to extract information on the loss rates in the system. At the same time, one should check if the number of polaritons at the condensation threshold is a constant, which I would expect for a close-to-equilibrium system, or if there is some dependence on the external parameters.

We agree that with longer pump pulses the population inversion in the system does not exceed the threshold of the condensation (and not even the threshold of the stimulated thermalization process). It is possible that the condensation threshold would be reached also with longer pulses if the pump fluence was increased so that the peak power stayed the same as for the short pulse. However, indeed this is not

experimentally feasible as the nanoparticle arrays start to get damaged with pump fluences larger than 10 mJ/cm².

We have moved the data of Figure S10 of the original supplementary material to the main text, see the new Figure 7. We have also improved the presentation of the data such that we now present 1) the line spectra at the condensation threshold for each case (if such was observed) and 2) the threshold curves until higher fluences than before. Previously we had plotted all the line spectra at a constant pump fluence (the condensation threshold of 50 fs and 100 fs pulses), but thanks to the insightful comment of the reviewer we realized that it is more reasonable to look for a condensation threshold from higher pump fluences for the longer pulses. As an outcome, we found out that at 4.5 mJ/cm² also the 250 fs pulse creates a thermal tail almost as linear as the 100 fs pulse (at 3.5 mJ/cm²). Based on fitting the tail of the observed line spectra to the MB distribution, we can conclude that condensation with a thermal tail is observed for 100 and 250 fs but not for longer pulses. We have added the resulting fit temperatures to the manuscript and the goodness of fit values to the Methods subsection "Fits to the Maxwell-Boltzmann distribution" (Supplementary Section S3 in the previous version. Some signatures of condensation can be observed in the evolution of real space intensity pattern and spectrum even for 350 fs pump pulse but the line spectrum never reaches a linear thermal tail (measured up to 10 mJ/cm²). A saturation of the output fluence can be seen beyond 6 mJ/cm². In addition, the new Figure 7c presents the thresholds for lasing and condensation, as well as the start of the incomplete stimulated thermalization regime, as a function of pulse duration (based on a suggestion of Reviewer 3). We have also added two new supplementary movies, Supplementary Movie 3 and 4, which show the real space and k -space measurement for the 500 fs pulse as a function of full range of pump fluences.

The number of emitted photons, as well as the threshold, is constant for 50fs and 100fs. Based on this one could then argue, as the reviewer suggests, that in this timescale the condensate behaves like a close-to-equilibrium system. However, we cannot make conclusions based on only two data points. For the 250 fs pump pulse, the threshold is already higher. The idea presented by the reviewer that the loss rates and the degree of equilibrium could be estimated from the condensation thresholds for different pump pulse durations is excellent, but at the moment we have only three data points for the condensate case which is not sufficient for such analysis. We are thankful for the reviewer for presenting this idea and will certainly pursue it in our future studies.

iii) The observed thresholds seem a bit ambiguous, as they are only valid for this exact system. As it is already shown that changing the pump pulse duration affect the observations (lasing or condensation) greatly, it would also help if the dependence of the observed double threshold behaviour to other experimental parameters such as pump geometry or lattice size is discussed.

The condensation is actually very sensitive to the pump geometry meaning the angle of the pump pulse hitting the array. The spatial intensity profile in all our experiments has been flat. To obtain a clear MB distribution of the tail in the time-integrated luminescence signal, the pump pulse needs to come at normal incidence, and even a slight misalignment changes both the real space intensity pattern and the spectral distribution. Non-zero incidence angle results in a time difference of the excitation at the two edges of the array (e.g., 10 degree angle means 60 fs time difference), and can cause an asymmetry in the populations of counter-propagating polariton modes.

Regarding the lattice size and other parameters, the condensation occurs in sweet spots of a multi-dimensional parameter space. The dye concentration affects the thermalization speed. The lattice period affects the band edge energy, and for a fixed pump energy it therefore affects the energy interval over which the polaritons need lose their energy in the thermalization process. Thus, one needs to match the lattice size (distance that the excitations need to propagate from the array edges to the centre), the dye molecule concentration (thermalization speed) and the lattice period (band edge energy) so that the number of thermalized polaritons is high enough to form a condensate at the band edge. Other parameters that have been varied in the experiments include the particle size and shape. It seems that for smaller particles the

scattering is not sufficient to create strong enough SLR modes and for larger particles (and for rod shaped ones) the excess metal that results in higher losses prevents the formation of the condensate, even though thermalization (Maxwell-Boltzmann distribution) may be present.

In the end of the "Effect of pulse duration" section of the original manuscript we had one sentence commenting the parameters. Now that part of the text has been replaced by a new paragraph which presents a more detailed discussion on the effect of the parameters.

iv) I am not convinced if distinguishing the stimulated thermalization regime so strongly is justified. From the observed energy distributions, this regime seems to be a regime of imperfect thermalisation prior to condensation? Or more concretely, what (experimental) criterion is used to call a spectrum thermalized? One could e.g. look at the deviation from a Boltzmann distribution and constrain this somehow to have a quantitative measure.

We agree that the intermediate regime presents imperfect thermalization prior to condensation. This was actually said on page 13 of the original manuscript ("At the intermediate regime of pump fluences, however, a thermal distribution is not reached before the population decays.") and on page 14 as well ("... an intermediate regime showing an incomplete stimulated thermalization process,..."). But indeed in the beginning of the manuscript our wordings may have given a misleading picture of the intermediate regime. We have now changed the words "stimulated thermalization" into "incomplete stimulated thermalization" in the abstract, changed "showing thermalization features" to "showing features of a thermalization process" in the introduction, and "thermalizing population of polaritons that propagate" to "polariton population undergoing a thermalization process and propagating" in the second paragraph of the Results section. We have also added a definition of how we distinguish the regime of the incomplete stimulated thermalization in the new Fig. 7 caption. The threshold for entering the incomplete thermalization regime was chosen to be the pump fluence at which the red shift and the associated interference pattern (such as in Fig. 2e) becomes visible in the real space spectra.

On the question what criterion is used to call a spectrum thermalized: the distribution is thermalized when it fits the MB distribution (at high energies), which in our system happens only in the condensate regime. In that regime, the tail of the distribution fits with the MB distribution with temperature error of ± 2 K (at 95% confidence level). "How well" the distribution is described by the MB distribution can be further quantified by the goodness of fit. The square root of the variance of the residuals (RMSE) and the R-square value for the fit are 107 and 0.996, respectively. The R-square value of 0.996 means that the fit explains 99.6% of the total variation in the data about the average. Whereas R-squared is a relative measure of fit, RMSE is an absolute measure of fit and indicates how close the observed data points are to the model's predicted values (in the same units as the data, in our case the number of counts which ranges up to 10^4 , so 107 is quite good).

We would like to note that to obtain the goodness of fit values we had to perform the fits again and this resulted in a slight change in the resulting temperatures and error limits. In the original version of the manuscript we had averaged the data over a number of points in energy. Now we have redone the fits for the raw data without averaging so that more data points are included in the fitting procedure. This makes the fit more stable and we actually obtain narrower error limits for the fit temperatures. The new (old) fit temperature with 95% confidence bounds for the distribution presented in manuscript Fig. 2a and Supplementary Fig. 6a is 313 ± 2 K (333 ± 12 K). For the two higher pump levels discussed in Methods (previously in Supplementary Section S3), the new (old) values are 282 ± 2 K (291 ± 7 K) and 250 ± 2 K (262 ± 9 K). We have updated the fit temperatures as well as added the goodness of fit values in the manuscript and in the Methods subsection "Fits to the Maxwell-Boltzmann distribution" (Supplementary Section S3 in the previous version). We also added there a sentence specifying which values of energies are included when fitting the MB distribution to the tail.

v) In the previous experiments of the research group a single-mode condensate was achieved. Thus, more discussion on the multimode behaviour in the condensed regime is also necessary. There is some information about possible explanations of this in the supporting file. However, the discussion will be much more clear if the simulated mode structure of the lattice is plotted beside the observed spectra. Also, are there any other possible explanations for the observed multimode behaviour? Why is it not simply multi-mode lasing? The definition of a multimode condensate is anyway to be taken with a grain of salt, as most people take macroscopic population of the ground state as "the condensate", thus multimode condensates in itself are not governed by this Definition, so some discussion is required.

We have checked that the multiple peaks do not originate from trivial optical effects such as Fabry-Perot interference, or finite lattice size in a straightforward "particle in the box" -sense. This was already mentioned in the original manuscript. We have also confirmed that the peaks do not correspond to waveguide modes in the sample [1, 2, 3]. This information has now been added to the manuscript.

References:

- [1] Hammer, M. 1-D multilayer slab waveguide mode solver. <https://www.computational-photonics.eu/oms.html>
- [2] Winkler et al. arXiv:2001.07594 (2020)
- [3] Schokker et al., Phys. Rev. B 90, 155452 (2014)

We have now done substantially better simulations of the lattice modes, and have a good qualitative understanding of the origin of the multiple peaks. We extended our T-matrix code to be able to handle complex frequencies instead of purely real. This gives us information of not only of the energy but also the lifetime of the eigenmodes. Furthermore, we performed simulations also for finite size lattices in addition to the infinite one, and we used cylindrical nanoparticles that correspond to the experiment, instead of the spherical ones used in the original manuscript. The text in Supplementary Note 3 (Section S6 in the previous version) has undergone major revision correspondingly. The new results are shown in a new Supplementary Figure 2. In the infinite lattice, three distinct eigenenergies appear at the Γ -point, while a large number of possible modes remain degenerate. This splitting into three modes comes from the finite size of the nanoparticles, as mentioned in the original manuscript, but is much more prominent for the cylindrical nanoparticles used here in contrast to the spherical ones of the original manuscript. In finite lattices, the discrete translational symmetry of the infinite lattice is broken, and the degeneracy is further lifted, a much larger number of distinct energies appears. The number of the modes and their energy separation depends on the size of the lattice in a complicated way. We have added text to the manuscript, on page 8, to mention this and to direct the reader to the new simulations and results of Supplementary Note 3 and Supplementary Figure 2 (Section S6 in the previous version).

In the simulations of the finite size system, the discrete translational symmetry of an infinite system cannot be utilized and they become much more demanding, the computational time increasing with the system size. Therefore we are, with our present computational resources and code, unfortunately not able to simulate arrays of the same sizes as used in the experiments. The conclusions that we can thus draw from the present simulations are qualitative, and we cannot present the simulated energies next to the observed ones since they are for different size systems. However, qualitatively it is clear that even an infinite lattice offers three distinct energies, and finite size lattices further ones, with energy splittings that are sensitive to the lattice size in a non-trivial way. The simulated energy splittings are of the same order of magnitude as the observed ones.

We do not know the reasons why only a single condensate peak was observed in our previous work. One factor that may contribute is that the lattice was of different size (300 μm in the propagation dimension) and the nanoparticles were rod-shaped, so the eigenenergies and their separations are different. Another factor is the larger number of excitations present in the present study, which might help to create multimode condensation.

It is true that in perfect thermal equilibrium only the lowest energy state gains macroscopic population. In driven-dissipative systems, however, condensation to several modes at distinct energies is possible. We have added on page 8-9 a sentence that mentions this, and given references (which contain further refer-

ences on the topic).

vi) It is interesting that by tuning the pump pulse duration and the lattice size, a spectral distribution that very closely matches Maxwell-Boltzmann distribution can be attained. I think the fact that this equilibrium distribution is achieved outside a steady state deserves more discussion, apart from the short discussing on page 14 and in the supplementary S5. While in a thermal equilibrium system the condensate is always achieved, here it seems more like an interplay between losses and gain, similar to the experiments on classical condensation by e.g. Baruch Fischer, and thus I would hope for a longer discussion on the physics. Additionally, some comments on which thermodynamic quantities are expected to follow predictions for thermal equilibrium in such a case apart from the spectrum would help.

Let us answer these points in several steps, first the classical vs. Bose-Einstein (BE) (-type) condensation, then the question why the BE distribution (Maxwell-Boltzmann tail) can be observed outside a steady state, and finally the thermodynamic quantities.

Classical vs. BE condensation.

The BE distribution $1/(\exp((E - \mu)/kT) - 1)$ can be expanded at high ($E - \mu > kT$) energies as $\exp(-(E - \mu)/kT)$ which is the classical thermal Maxwell-Boltzmann (MB) distribution. In a linear scale, the MB distribution grows towards low energies (has a “peak”), so at a glance one might not be able to distinguish it from a BE distribution. However, in log scale, the MB distribution is a straight line at all energies. We show that our data plotted in log scale has the expected linear MB behaviour at high energies, but notably also a strong additional peaked feature at low energy. Thus the data clearly deviates from the case of classical MB distribution *only*, and is better described by the BE distribution.

The low energy ($E - \mu < kT$) expansion of the BE distribution is $kT/(E - \mu)$. So-called classical condensation of light or other waves (also called Rayleigh-Jeans (RJ) condensation) is a phenomenon where randomness combined with interactions between light modes and suitable mode-dependent gain/loss profiles can produce a distribution of the form $A/(E - B)$ (e.g., Sun et al., Nat Phys 8, 470 (2012), Fisher and Weill, Optics Express 20, 26704 (2012), Fischer and Bekker, Optics and Photonics News, September 2013, p. 40 (2013), Oren et al., Optica 3, 145 (2014), Ruckriegel and Kopietz, PRL 115, 157203 (2015); and references therein). Here A and B are system parameters and can be formally associated with temperature and chemical potential so that the distribution mimics the BE distribution at low energies. Note, however, that for instance A is related to the properties of the noise but not directly to some physical temperature T of a thermal bath. As $kT/(E - \mu)$ is not linear in log scale, the long linear (in log scale) tail of our distribution clearly shows that the concept of classical (RJ) condensation cannot describe our observations. We have confirmed by fits of our data to a function of the form $-\log(E - \mu)$ that we definitely do not have the distribution expected for RJ condensation.

Furthermore, our systems do not have the conditions typically needed for the classical (wave) condensation phenomena. In classical condensation, the randomness is given for instance by white noise or purposefully introduced random-phase fields, or spontaneous emission. We do not have the first two or any other known randomness source, and as the phenomenon we see is strongly stimulated, spontaneous emission noise is likely to be negligible. The interactions between the light modes in classical condensation are given for instance by four-wave mixing or temporal modulation in active mode locking. In our case, the light modes interact via the emission and absorption processes provided by the molecules. The losses of our modes are constant at the linear part of the dispersion, not mode-dependent as in some models of classical condensation. Now, one might still argue that, although to our best knowledge we are far from the conditions of classical condensation, maybe by accident we are there. For this reason, it is important that our observed distribution clearly rules out the possibility of classical (RJ) condensation. Note that the tail of the distribution has to be long enough, that is, $E - \mu$ has to be large enough compared to kT , to be able to make this conclusion. This is because at small energies ($E - \mu < kT$) the BE and RJ condensation resemble each other. Here $E - \mu$ means the deviation in energy from the condensate peak position. We show

a prominent (signal much bigger than noise) linear tail in an energy range of 75 meV, which is three times the room temperature $kT = 25$ meV, thus the range is large enough to rule out RJ condensation. In our previous work (Hakala et al. Nat Phys 14, 739 (2018)), the data (above noise level) ranged only 20 meV, so the present manuscript presents important additional evidence for BE-type condensation as compared to the RJ one.

In summary, our observed photon distribution deviates from both the classical (RJ) condensation and the classical (MB) thermal distributions, but agrees with the BE distribution which has a MB tail combined with large population at low energy. We have added a discussion on this matter in a new paragraph (starting "We attribute the red shift", page 14), together with some of the above mentioned references and a reference to Kirton, Keeling PRA 91, 033826 (2015) which presents an instructive discussion about the differences of the RJ and BE condensation as well as lasing. Note: The reviewer mentioned "experiments on classical condensation... by Baruch Fisher"; we have added relevant publications to the manuscript. We also added a citation to a recent paper by the B. Fisher group where they report a BEC of photons in an Erbium fiber (Weill et al., Nat Comm, 10, 747 (2019)).

Why MB distribution out of steady state

The conditions in our system are close to that of a photon BEC: we use organic molecules and absorption-emission processes for the thermalization. The main differences, however, are that 1) our excitations are not purely photonic but contain a plasmonic part, 2) we are at the strong coupling regime of light-matter coupling, and 3) all time-scales are orders of magnitude shorter. Theory (e.g. the works by Kirton and Keeling, Carusotto, and others) and experiments (by the Weitz group, R. Nyman, and others) have established that the photon BEC is distinct from both from the classical condensation discussed above and from lasing. These works show that, in a steady state, the BEC phenomenon occurs (and a MB tail is formed) when the thermalization process is sufficiently fast compared to the losses of the system (that is, emission and absorption cycles can take place within the life-time of the system). We have used the Kirton and Keeling theory to show that the MB tail forms also for pulsed excitation, please see the simulations in Hakala et al. Nat Phys 14, 739 (2018) (the theory has been used with pulsed pumping also in Walker et al., arXiv:1908.05568v1, (2019), <https://arxiv.org/abs/1908.05568>). Thus, in the weak coupling regime, the explanation is clear: subsequent emission-absorption processes, provided by molecules whose emission and absorption follow the Kennard-Stepanov relation, produce a BEC with thermalized distribution, both in steady state and pulsed situations.

In the experiments of the present manuscript, we are at the strong coupling regime, and a theory that can fully describe the situation has not been published to date. We can only speculate that the process is similar, although probably not identical, to the one that we have observed in a similar system at the weak coupling regime (and which is well described by the Kirton-Keeling weak coupling photon BEC theory). That is, we can argue (although not prove) that the emission-absorption processes lead to the room temperature MB distribution also in our present strong coupling case.

We can experimentally show that the thermalization process has a strongly stimulated nature. This makes it so fast that it occurs within the lifetime of the modes and thus fulfills the condition set to condensation by the weak coupling theory. Whether this explanation fully holds in the strong coupling regime is an interesting question which we expect the near-future theory developments to answer. We believe our results are highly inspiring and valuable in this sense: providing a challenge to the theory rather than a fully explained observation. Therefore we try to avoid over-interpreting the results, as also recommended by the referee.

We have added to the revised manuscript (see the three new paragraphs in the end of the Section "Stimulated nature of the thermalization") a discussion that explains the formation of MB distribution in an out-of-steady-state situation in the weak coupling case, and a suggestion (supported by our toy-model simulations) that similar phenomena may take place in the strong coupling case.

Which thermodynamic quantities can be observed

We have already extracted the temperature T from the tail of the distribution. Thermodynamic quantities such as energy, pressure, specific heat and entropy can be deduced from the observed photon energy distributions, along the lines presented in a work by the Weitz group (also mentioned by the referee below): Damm et al. Nat Comm 7, 11340 (2016) (<https://www.nature.com/articles/ncomms11340>). The photon distribution enters the formulas for various thermodynamic quantities, which are then given as function of T/T_c . Note that the actual temperature T was not varied in Damm et al. Instead, the photon number is varied (this is easily done by changing the pump fluence), which changes T_c and consequently T/T_c . Evaluating thermodynamic quantities from the distribution is an interesting future project and will reveal whether they follow equilibrium predictions, as in Damm et al. We have added to the manuscript the sentence "For instance, thermodynamic quantities can be determined using the observed photon distribution" together with a citation to Damm et al.

vii) The conclusions contain claims that are not supported I believe: The authors make several claims that have already been addressed for photon condensates, mostly by the Weitz group: - "What is the nature of the energy relaxation, how does the condensate form" - > Ref. 32 - "What are the quantum statistical properties" - > Ref. 21 and Nature communications 7, 11340 (2016) - "long-range coherence properties" - > PRL 116, 033604 (2016), Nature communications 8 (1), 158 (2017) Thus, these claims are not really supported in general. Of course they are still true for the polariton-dye system, but not for photonic BECs per se.

Certainly, these issues have been addressed for photon condensates, and also excessively for inorganic polariton condensates. For both systems, there are still many things to fully understand, but we agree that we should not give the impression that the above questions have not been addressed at all. These questions are to a much larger extend open (although already addressed in a few studies) in case of room temperature, strongly coupled condensates where the condensate dynamics is in the sub-picosecond regime. These strongly coupled, room temperature, ultrafast condensates include the system in our present manuscript as well as some organic polariton condensates. We have changed in the conclusions the words "These questions become challenging to answer for room temperature condensates..." to "These questions become challenging to answer for strongly coupled room temperature condensates..." This now excludes the photon condensates studied by the Weitz group (and others) since they are at the weak coupling regime. We have added references to our manuscript so that all the above mentioned articles are now cited.

viii) The abstract contains the claim "Our results give us direct access to spatiotemporal control and monitoring of thermalization processes...". While I agree with the control, which is here obtained by changing the pulse length, the "monitoring" part is to my opinion not shown in the paper. While for the propagating system as in previous work of the Törmä group a direct matching between position and time can be made, here a temporal monitoring would require a very sophisticated solution. There are streak cameras with 100 fs resolution available, but even this resolution obviously does not suffice.

We agree that our present experimental configuration does not give direct access to the temporal monitoring in the same sense as our previous work. However, it does give quite interesting indirect information about the dynamics. For instance, the low intensity areas in the arrays (see Figure 5) and the dependence of their size on pump fluence (and independence of the lattice size) was the key to reveal the stimulated nature of the redshift process. The width of the low-intensity area divided by the group velocity of the lattice dispersion gives an estimate for the time scale of the pulse build-up process. The width of the low-intensity areas depend on pump fluence as shown in Fig. 5i. For example, for the cases shown in Fig. 5a-f, at pump fluence of 2.2 mJ/cm^2 , the width is $25 \mu\text{m}$ which converts to around 175 fs. However, we agree it is better not to oversell this and the other possibilities of monitoring that our system offers. We have taken the words "direct" and "spatiotemporal" away from the abstract, the sentence now reads "Our results give access to control and monitoring of thermalization processes and condensate formation at sub-picosecond timescale."

Thus, in general I think the results of the paper are very interesting and deserve publication. However, in the current version of the manuscript the quite general claims made are not supported by the data, or not explained such that the reader could easily connect the data to the claims. Correspondingly, for publication in nature communications major revisions are needed to my belief. Most importantly, a clear explanation of the thermalization process, see my point (i), is required, as only then the reader can understand the interplay of the different mechanisms involved.

Correspondingly, the paper can only be accepted after major revisions, or should be sent to a more specialized Journal after revisions are made.

We thank the reviewer again for comments that helped us considerably improve the manuscript. We have made a major revision: We added several new paragraphs of explanations so that the connection between the data, the claims and related work in the literature becomes more clear. In particular, we now explain the thermalization process in more detail, as requested in point (i), in the new paragraphs. We have added new T-matrix simulations in the Supplementary information, moved material related to the original Figure S10 (intermediate pulse durations) from Supplementary information to the main text and presented it in a new way (new Fig. 7), added a number of new references, more accurate definitions and data fits, clarifications of various points as well as substantial amount of additional information that the three reviewers asked. Additionally, following the editor's request to fulfill the Nature Communications format guidelines, we have restructured the manuscript by moving a large part of the supplementary material into a new Methods section in the end of the revised manuscript. We hope the manuscript is now suited for publication in Nature Communications.

Reviewer: 2

The manuscript "Sub-picosecond thermalization dynamics in condensation of strongly coupled lattice plasmons" by Väkeväinen et al. demonstrates ultrafast thermalization dynamics of polariton lasing and condensation in a plasmonic nanoparticle array with strong light-matter coupling. In particular, the authors find three distinct regimes of light emission separated by two thresholds, with the first threshold corresponding to a one-dimensional lasing regime and the second threshold, which is only observed for short enough pulse durations, assigned to a two-dimensional multimode condensation regime including thermalization. These results are very interesting and present a novel phenomenon that to my knowledge has not been observed in these or similar systems before. The work thoroughly explores the different regimes and provides rich insight into the properties and dynamics of the emitted light. I thus wholeheartedly recommend publication in Nature Communications, provided the following comments are addressed:

We thank reviewer for appreciating our results and for the insightful comments which we answer below.

1) The authors mention that they achieve a signal that is 5 orders of magnitude larger than in their previous Nature Physics paper (ref. 28), which enables the observation of a thermal tail after condensation. However, it is not clear to me how or why this is achieved and what is the crucial difference with respect to the previous setup.

First, there are two factors that contribute to having more excitations in the system: less photo bleaching, and different pumping and detection geometry. Second, since stimulated processes are going on, the increased amount of excitations lead to an output emission that is enhanced in a nonlinear manner. In other words, increased amount of initial excitations, together with nonlinear stimulated effects, lead to the five orders of magnitude larger signal.

The dye molecule film in the present case is orders of magnitude thicker than in our previous experiments (1 mm vs. $\sim 10 \mu\text{m}$). This makes the sample very robust and photo bleaching is not an issue during

the experiment. When we vary the pump fluence, there is a few seconds break between consecutive measurements with different pump fluence values. During that time, natural circulation of the dye molecules in their solvent replaces any bleached molecules. In the Nature Physics paper samples, this type of replenishment was not efficient, and we had to stop the experiments at a certain pump fluence because the sample was bleached. In the present case, we were able to use higher pump fluences.

There is a major difference in the pumping and luminescence collection geometry. In the Nature Physics article, the molecules were excited at one end of an elongated nanoparticle array and the pump spot covered less than 10% of the array. The excited molecules emitted photons to propagating SLR modes, and the propagation took place in part of the array where there were only ground state molecules. When evaluating the number of photons in the condensate, we collected data only from part of the array, namely the part where the BEC had formed after propagation and thermalization of the excitations. Photons from other parts of the array were not counted because there the system had not yet reached the BE-distribution and therefore those photons were not part of the condensate. During the propagation and thermalization, excitations were lost in both radiative and non-radiative channels. In the present case, the pump spot covers the whole array, and all the propagation of excitations takes place in an area where there are excited molecules. We also collect the photoluminescence from the whole array.

The differences in the geometry of pumping (partially overlapping with an edge vs. covering the whole array) and collection (part of the array vs. whole array) would as such increase the signal only in a linear manner. They cannot, however, be accounted for more than an order of magnitude or two increase. Non-linear effects, namely the stimulated thermalization, can explain the five orders of magnitude increase. The increased amount of excitations leads to a stimulated thermalization and condensation process which makes the excitations couple out as light in a very short time instead of decaying into other loss channels of the system.

We have now added in the revised manuscript, page 7-8, a sentence commenting these points and referring to Supplementary Note 2 where we have added a detailed discussion on them.

2) It is also not fully explained why the authors refer to the regime reached after the first threshold as lasing and after the second as condensation - this seems to rely on the observation of a thermalized tail after the second threshold, but the reasoning behind this should be clarified.

Yes, we refer the second threshold as condensation due to the existence of a thermalized tail since this is a feature of a BEC-type phenomenon. The first threshold is not associated with a thermalized tail, so we refer to it as lasing since the characteristics are similar to lasing observed previously in nanoparticle arrays. We are at the strong coupling regime, however, so the lasing can be called polariton lasing as well. Polariton lasing, on the other hand, is sometimes referred to as polariton condensation in the literature. So clarification is needed. We have now explained in more detail the reasoning behind the names used (see the new text starting "We use the existence of a long..." in the end of the paragraph that starts "The sample luminescence", page 6).

Note that another difference between the two regimes is that the first threshold leads to 1D spatial coherence and the second one to 2D coherence. This, however, is not the reason for the names, since also lasing in nanoparticle arrays can under some conditions show 2D coherence. The different coherence properties, however, highlight that the phenomena seen at the two thresholds are different.

3) The Michelson interferometer measurements seem very similar to those recently reported in reference 30, which also showed the stripes observed in the "1D lasing" regime identified here. It would be helpful to a reader not familiar with the literature to mention this. Also, could further insight be provided into why these 1D stripes show up and full 2D coherence is not reached in this regime?

The Michelson interferometer experiment that we do is a standard technique that has been used widely in the literature for measuring spatial coherence of various different types of lasing and condensate systems,

and is indeed used also in Ref.30. We are not sure which stripes the referee refers to, and this made us to realize that we have to clarify our description of the Michelson experiment because there are actually two types of stripes visible in our images: stripes that arise in this type of Michelson experiment for any coherent (lasing, condensation, etc.) system, and other stripes that are specific to nanoparticle arrays. These are explained below.

First about stripes specific to nanoparticle arrays: In our case one-dimensional lasing originates from linearly polarized dipolar nanoparticles. The nanoparticles are polarized in x-direction and predominantly radiate in y-direction. The feedback is provided by counter-propagating optical modes in y-direction, and long-range radiative coupling in x-direction is not efficient. This produces one-dimensional vertical stripes in the real space images (see Figs. 4b and 4d) (these can be understood as individual (or just a few neighbouring) nanoparticle chains lasing independently). Due to the one directional coupling, one dimensional lasing shows high spatial coherence only in the direction of the feedback (y-direction). Please see also the sum of non-interfered real space images in Supplementary Figure 3e-h for reference.

Then about stripes related to the Michelson interference: Horizontal stripes in Fig. 4b are interference fringes that arise from overlapping two real space images, one of which is flipped with respect to the x-axis, at the camera sensor in the Michelson interferometer setup. There horizontal stripes (fringes) occur on top of the vertical lasing stripes. In Fig. 4d, the flipping is done with respect to the y-axis and since there is almost no spatial coherence in the x-direction, no additional fringes are obtained (confirmed by an additional Fourier analysis on the amplitude of spatial frequencies, explained in Supplementary Note 4 (Section S7 in the previous version).

In the condensation regime, there is a more uniform intensity pattern visible in the central part of the array and the Michelson interferometer produces the interference fringes in both x- and y-direction (see Figs. 4c and 4e). This is in contrast to the lasing regime that shows the interference fringes only in the y-direction. However, we have noticed that by increasing the size of the nanoparticles, two dimensional spatial coherence can be obtained also in the lasing regime, due to multipolar excitation of individual nanoparticles (Ref. 30 used multipolar nanoparticles). However, the nanoparticles in our current experiment ($d = 100$ nm) are too small (with respect to the wavelength range of interest) to exhibit 2D coherence in the lasing regime.

Finally, it is important to note that the Michelson interference fringes occur with a fixed period determined by the experimental setup (incoming angle of the overlapping images), and therefore these fringes can be distinguished from any other stripes in the real space images (with different period). This has been utilized in the Fourier analysis of spatial frequencies explained in Supplementary Note 4 (Section S7 in the previous version).

We have added a sentence explaining the various stripes in the end of the caption of Figure 4, and further explanation along the lines given above in the Supplementary Note 4, including a reference to Ref. 30 (it is also cited in the manuscript) of the revised version).

4) The continuous redshift of the emission during the propagation implies a somewhat continuous thermalization process, which seems slightly at odds with the picture of (resonant) vibration-driven decay between states that is often invoked in organic molecular polariton condensation. For example, this is identified in a very recent article (Ramezani et al, Nano Letters 2019, <https://doi.org/10.1021/acs.nanolett.9b03139>, which should probably be cited) as the main pathway for polariton decay, also on ultrafast timescales. Can it be understood where this seeming difference between decay driven by discrete vibrational modes and continuous redshifting comes from?

The “continuous redshift” and “decay by discrete vibrational modes” are different regimes of the same phenomenon. The microscopic description of both consists of, in its simplest form, an electronic transition, vibrational states (coupled to outside degrees of freedom, for instance a thermal bath), and several light modes, all of these coupled with each other. With different approximations, different phenomena can

be derived from this starting point. The Kirton-Keeling model of photon condensation (Kirton, Keeling, PRA 91, 033826 (2015)) leads to a continuous redshift, similar to the one observed by us here, which is needed for producing the condensate with a MB tail. Interestingly (see, e.g., figs. 2 and 6 and related discussion in Kirton, Keeling, PRA 91, 033826 (2015)), if the vibrational level structure is hardly visible in the absorption and emission spectra, a thermalized BEC is easier to achieve, while in case of prominent vibrational shoulder structure in the spectra, one needs longer cavity lifetimes for it. Thus in the case where a vibrational level structure is prominently visible in the abs/em spectra and the cavity lifetime is short, the resulting relaxation process may better be described by the “vibrationally assisted relaxation” picture, as often done in the context of organic polariton condensates. This picture is described for instance in the early works of La Rocca and coworkers: in PRB 80, 235314 (2009) they calculate rates for polariton scattering mediated by photon emission and creation of vibrational quanta. Such scattering rates (Fermi’s Golden rule/linear response) contain delta-functions that enforce energy conservation (namely, that the energy of an initial polariton (or exciton) equals the energy of the final polariton plus a vibrational quantum), leading to “discrete” relaxation to the condensate. In reality, the delta-function is broadened by the line-widths of the states involved. Now if one assumes that such broadening would be large compared to the vibrational level spacing, one gets back to a continuous process.

For our molecules, vibrational shoulders are not visible in the absorption and emission spectra, see Fig.1d of the manuscript. In the Ramezani et al. article that the reviewer mentions, in contrast, the absorption and emission spectra (Fig.1a of that article) show clear vibrational substructure.

Our toy model simulations in Supplementary Figure 5 and Supplementary Note 6 (previously Section S9) also illustrate this point. In Supplementary Figure 5b,e,h, the green curve corresponds to the mode for which the energy conservation condition mentioned above is fulfilled. Indeed that mode is somewhat favoured, but the other modes do get prominent population as well, because of finite line-widths and since strong coupling produces effective line-broadening that allows population at non-resonant states.

Discussion on this issue has been now added to the manuscript, in a new paragraph (the last paragraph of the “Stimulated nature of the thermalization” section), as well as a reference to the article by Ramezani et al.

5) After the second threshold, the “condensate” peak has a multi-peak substructure in energy (see, e.g., Fig 3j, lowest panel). This is explained in the supplemental (section 6) as due to an approximate degeneracy between different modes that is lifted for finite-sized nanoparticles. However, it did not become clear to me what these different modes correspond to, maybe this could be clarified?

Reviewer 1 asked a related question (item v)), so we use part of the text from that answer here.

We have now done substantially better simulations of the lattice modes, and have a very good qualitative understanding of the origin of the multiple peaks. We extended our T-matrix code to be able to handle complex frequencies instead of purely real. This gives us information of not only of the energy but also the lifetime of the eigenmodes. Furthermore, we performed simulations also for finite size lattices in addition to the infinite one, and we used cylindrical nanoparticles that correspond to the experiment, instead of the spherical ones used in the original manuscript. The text in Supplementary Note 3 (Section S6 in the previous version) has been updated correspondingly. The new results are shown in a new Supplementary Figure 2. In the infinite lattice, three distinct eigenenergies appear at the Γ -point, while a large number of possible modes remain degenerate. This splitting into three modes comes from the finite size of the nanoparticles, as mentioned in the original manuscript, but is much more prominent for the cylindrical nanoparticles used here in contrast to the spherical ones of the original manuscript. In finite lattices, the discrete translational symmetry of the infinite lattice is broken, and the degeneracy is further lifted, a much larger number of distinct energies appears. The number of the modes and their energy separation depends on the size of the lattice in a complicated way.

We have added text to the manuscript, on page 8, to mention this and to direct the reader to the new simulations and results in Supplementary Figure 2 and Supplementary Note 3 (Section S6 in the previous

version).

In the simulations of the finite size system, the discrete translational symmetry of an infinite system cannot be utilized and they become much more demanding, the computational time increasing with the system size. Therefore we are, with our present computational resources and code, unfortunately not able to simulate arrays of the same sizes as used in the experiments. However, qualitatively it is clear that even an infinite lattice offers three distinct energies, and finite size lattices further ones, with energy splittings that are sensitive to the lattice size in a non-trivial way. The simulated energy splittings are of the same order of magnitude as the observed ones.

We thank the reviewer again for comments that helped to considerably improve our manuscript. After these changes and those requested by other reviewers, as well as format changes to follow the journal style (much of previous Supplementary information is now in Methods), we hope the manuscript is now ready to be published in Nature Communications.

Reviewer: 3

The manuscript reports about an experimental observation of the sub-picosecond thermalization dynamics in the condensation formation of strongly coupled lattice plasmons. The high quality and the level of detail of the obtained results as well as their concise presentation justify that the manuscript is published at Nature Communications. However, prior to publication, the authors should respond to the following questions and comments:

We thank the reviewer for the positive evaluation of our work and for the comments raising up important points. Please see below our answers and explanations of the revisions done.

1) Is it possible to estimate the magnitude of the photon-photon interaction and to determine at least qualitatively how it depends upon the concentration of the dye molecules?

For photon condensates, there are known approaches to determine the effective photon-photon interaction (e.g., Stein et al., New J. Phys. 21 103044 (2019), Klaers et al., Nature 468 545-548 (2010), Klaers et al., Appl. Phys. B 105 17-33 (2011), Marelic et al., Phys. Rev. A 94 063812 (2016), Radonjic et al., New J. Phys. 20, 055014 (2018), Stein et al., New J. Phys. 21 103044 (2019)). Because the thermalization mechanism in our case is similar to photon condensates, we might consider using such approach. But we are at the strong coupling regime, therefore we should probably talk about polariton-polariton interactions, and approach such effective interactions in a way similar to organic polariton condensates, as discussed in the following paragraph (which is directly copied from the response to Reviewer 1 who asked a similar question).

Polariton-polariton interaction can originate from Coulomb interaction of the excitonic part (the molecular part) of the polariton, and such repulsive interaction would show up as a blueshift of the condensate energy with increasing pump fluence (increasing number of polaritons). Also polariton - reservoir exciton interactions can contribute to the blueshift. On the other hand, an effective interaction can arise also from saturation effects, in other words, when the system has a higher number of excitations, the collective light-matter interaction decreases; such effects can be theoretically shown to lead to an effective repulsion between polaritons (and between polaritons and reservoir excitons), which also shows up as a blueshift. Extracting the interaction constant from the observed blueshift is thus tricky, since various types of Coulomb and saturation effects contribute, moreover, the functional dependence of the blueshift on the polariton number can be complicated. However, one may make a mean-field approximation estimate for low polariton density (linear regime) (Deng et al., Reviews of Modern Physics 82, 2010). In this case the blueshift ΔE is linearly dependent on the polariton density n and the interaction constant g , $\Delta E = gn$.

Putting in our experimentally observed values of $\Delta E \sim 20$ meV and 10^{17} polaritons/m² we obtain a value of $g \sim 0.2\mu\text{eV}\mu\text{m}^2$. We believe most of the effective interaction in our case is due to the saturation effects, that is, degradation of the light-matter coupling, since the whole dispersion blue-shifts. We expect the Coulombic effects are of a much smaller order of magnitude, as is typical in organic semiconductor polariton condensates. For instance, in the organic polariton condensate of Daskalakis et al. Phys. Rev. Lett. 115, 035301 (2015), the Coulomb interaction -caused polariton-polariton interaction was estimated to be $\sim 1\text{-}10\mu\text{eV}\mu\text{m}^2$ (e.g., Fig. 5 in Estrecho et al., Phys. Rev. B 100, 035306 (2019)). Further analysing the blueshift, taking into account all the above mentioned complexities, and understanding the nature of interactions in this system is beyond the scope of the present manuscript. An interested reader can easily calculate the $g = \Delta E/n$ from the numbers we give in the revised manuscript (Fig.2 gives the number of photons observed from the $100\ \mu\text{m} \times 100\ \mu\text{m}$ sample, which reveals n , and the blueshift can be obtained from the numbers given in the paragraph "In Fig.4, ...").

We did not observe the BEC for concentrations lower than the 80 mM. Indeed it would be interesting to increase the concentration in small steps starting from the 80 mM one, and observe how the blueshift depends on concentration. To relate that to effective interactions is then a subtle task, as explained above. We view that this is a key future research topic but it is beyond the scope of the present work.

2) *Which criteria justify to choose the particular organic dye molecule IR-792? Which other dye solutions could, in principle, be used?*

This is an excellent question. We have tried multiple different dye molecules and the two main criteria for a good emitter for our experiments are 1) solubility in high concentrations (up to 100 mM to achieve strong coupling) and 2) high stability under optical excitation (slow photobleaching). A list of tested dye molecules and collected information are now given in Supplementary Table 1, which also explains the choice of the dye molecule IR-792.

3) *On the one hand, at the beginning the authors mention the topic of Bose-Einstein condensation of photons, but then on page 7 they argue to have measured only a Maxwell-Boltzmann distribution. Thus, the general question arises whether the observed condensation of strongly coupled lattice plasmons is a quantum or a classical effect. This question reminds one to a similar situation, which occurs for the condensation of magnons, where the original experimental publication*

Nature 443, 430 (2006)

claims to have seen a Bose-Einstein condensation, but later on this was more precisely identified to be just a Rayleigh-Jeans condensation in

Phys. Rev. Lett. 115, 157203 (2015)

Reviewer 1 (item vi) of the report) presented a related question, so we will answer with partly the same text here, with some further comments added.

The BE distribution $1/(\exp((E - \mu)/kT) - 1)$ can be expanded at high ($E - \mu > kT$) energies as $\exp(-(E - \mu)/kT)$ which is the classical thermal Maxwell-Boltzmann (MB) distribution. In a linear scale, the MB distribution grows towards low energies (has a "peak"), so at a glance one might not be able to distinguish it from a BE distribution. However, in log scale, the MB distribution is a straight line at all energies. We show that our data plotted in log scale has the expected linear MB behaviour at high energies, but notably also a strong additional peaked feature at low energy. Thus the data clearly deviates from the case of classical MB distribution *only*, and is better described by the BE distribution. In other words, it is not really that we measure "only Maxwell-Boltzmann distribution": we show that the tail of the distribution fits the MB distribution, but that there is also a prominent population at low energies that does not fit it. It is true that we did not emphasize this in the manuscript, this has been corrected now by adding a whole new paragraph (the third last paragraph of Section "Stimulated nature of the thermalization") explaining this and the difference to Rayleigh-Jeans condensation.

The low energy ($E-\mu < kT$) expansion of the BE distribution is $kT/(E-\mu)$. So-called classical condensation of light or other waves (also called Rayleigh-Jeans (RJ) condensation) is a phenomenon where randomness combined with interactions between light modes and suitable mode-dependent gain/loss profiles can produce a distribution of the form $A/(E-B)$ (e.g., Sun et al., Nat Phys 8, 470 (2012), Fisher and Weill, Optics Express 20, 26704 (2012), Fischer and Bekker, Optics and Photonics News, September 2013, p. 40 (2013), Oren et al., Optica 3, 145 (2014), Ruckriegel and Kopietz, PRL 115, 157203 (2015) (the article the reviewer mentions); and references therein). Here A and B are system parameters and can be formally associated with temperature and chemical potential so that the distribution mimics the BE distribution at low energies. Note, however, that for instance A is related to the properties of the noise but not directly to some physical temperature T of a thermal bath. As $kT/(E-\mu)$ is not linear in log scale, the long linear (in log scale) tail of our distribution clearly shows that the concept of classical (RJ) condensation cannot describe our observations. We have confirmed by fits of our data to a function of the form $1/(E-\mu)$ that we definitely do not have the distribution expected for RJ condensation.

Furthermore, our systems do not have the conditions typically needed for the classical (wave) condensation phenomena. In classical condensation, the randomness is given for instance by white noise or purposefully introduced random-phase fields, or spontaneous emission. We do not have the first two or any other known randomness source, and as the phenomenon we see is strongly stimulated, spontaneous emission noise is likely to be negligible. The interactions between the light modes in classical condensation are given for instance by four-wave mixing or temporal modulation in active mode locking. In our case, the light modes interact via the emission and absorption processes provided by the molecules. The losses of our modes are constant at the linear part of the dispersion, not mode-dependent as in some models of classical condensation. Now, one might still argue that, although to our best knowledge we are far from the conditions of classical condensation, maybe by accident we are there. For this reason, it is important that our observed distribution clearly rules out the possibility of classical (RJ) condensation. Note that the tail of the distribution has to be long enough, that is, $E-\mu$ has to be large enough compared to kT , to be able to make this conclusion. This is because at small energies ($E-\mu < kT$) the BE and RJ condensation resemble each other. Here $E-\mu$ means the deviation in energy from the condensate peak position. We show a prominent (signal much bigger than noise) linear tail in an energy range of 75 meV, which is three times the room temperature $kT = 25$ meV, thus the range is large enough to rule out RJ condensation. In our previous work (Hakala et al. Nat Phys 14, 739 (2018)), the data (above noise level) ranged only 20 meV, so the present manuscript presents important additional evidence for BE-type condensation as compared to the RJ one.

In summary, our observed photon distribution deviates from both the classical (RJ) condensation and the classical (MB) thermal distributions, but agrees with the BE distribution which has a MB tail combined with large population at low energy. We have added a discussion on this matter in the new paragraph mentioned above, together with some of the above mentioned references as well as a reference to Kirton, Keeling PRA 91, 033826 (2015) which presents an instructive discussion about the differences of the RJ and BE condensation as well as lasing.

4) *Another question concerns symmetry arguments. From Fig. S1 the nanoparticle array is symmetric in x - and y -direction. On the other hand*

a) *it is argued on page 9 that lasing exhibits confined luminescence along ky but spreads along kx*

b) *Fig. 4 reveals that spatial coherence is asymmetric in x - and y -direction.*

What is the physical reason for that asymmetry?

The system is as described in the manuscript and in the Methods subsection "System" (Supplementary Section S1 in the previous version): the period in y - and x -direction is $p_y = 570$ nm and $p_x = 620$ nm, respectively. As we explain in Methods, this is done because asymmetric periodicity separates the diffracted orders in the energy spectrum for the two orthogonal polarizations (e_x and e_y), and the SLR dispersions are correspondingly separated, which simplifies analyzing the measurement data. We understand that the

50 nm difference is so small compared to the periods, that by looking at the SEM image in Supplementary Fig. 8 (previously Fig. S1) the array might appear symmetric. We also did not realize that actually the SEM image is taken so that the array appears 90 degrees rotated. We have now clarified the issue by adding coordinate axes in the SEM image in Supplementary Fig. 8 (previously Fig. S1).

What comes to the physical reason, one-dimensional lasing in general originates from linearly polarized dipolar nanoparticles. If the nanoparticles are polarized in x-direction they predominantly radiate in y-direction. The feedback is provided by counter-propagating optical modes in y-direction, and long-range radiative coupling in x-direction is not efficient. Due to the one directional coupling, one dimensional lasing is confined in k_y (and not that much in k_x) and shows high spatial coherence only in the direction of the feedback (y-direction). Note that if the coupling in the x-direction is made strong as well, for instance by using multipolar nanoparticles, also 2D lasing could be achieved.

In most cases of nanoparticle array lasing reported in the literature, the polarization of the 1D lasing follows that of the pump beam. Various factors can contribute to this, for instance: 1) The pump excites molecules with dipole orientation parallel to the polarization and this orientation is perhaps not completely lost before the molecule emits. 2) The pump excites the single nanoparticle resonances of the individual particles and the resulting localized surface plasmon excitations have the same polarization as the pump. Molecules located in the field hot spots of the nanoparticles are excited more efficiently and they preferably emit to the corresponding polarization mode (this was suggested for instance in Ref. 41).

Furthermore, since our system is strongly coupled, the molecule and nanoparticle excitations are hybridized and have the same polarization. Either one of them or both are initially affected by the pump polarization, and thus we observe 1D lasing in the direction determined by the pump polarization. We have now added a few clarifying words in the caption of Fig. 4, and more text in Supplementary Note 4 (Section S7 in the previous version), to explain these issues.

5) The authors should mention in the text the order of magnitude of the coherence length and how to explain at least the order of magnitude.

The coherence length in the samples without pumping is $24\mu\text{m}$. We have now added a mentioning of the coherence length to the beginning of the Section "System". In the condensate, the coherence reaches over the whole sample for all sizes (the largest was $100\mu\text{m} \times 100\mu\text{m}$). The samples are therefore too small to extract information about the decay of spatial coherence; this is an interesting future research topic, but requires finding the parameters that enable observing the condensate also in large nanoparticle arrays. In textbook equilibrium 2D BECs, spatial coherence decays algebraically. But it has been theoretically predicted that the situation is more complicated in driven-dissipative condensates. Proper theory of spatial coherence properties of the type of ultrafast, non-steady-state, strongly coupled condensates as we have does not exist. We prefer not to speculate on the order of magnitude of the expected coherence. But we have added text that gives information about the observed coherence, and refers to the complexity of the question of spatial coherence in driven-dissipative systems: "The coherence both in the lasing and condensation cases extends over the whole array, for all array sizes (the largest was $100\mu\text{m} \times 100\mu\text{m}$), thus the coherence length is at least four times larger than that of the samples without pumping ($24\mu\text{m}$). In a future study, larger samples should be used for finding how the coherence decays (e.g., exponential, polynomial). Whether algebraically decaying phase order exists in 2D driven-dissipative systems is a subtle question (1,2, 45, 46)." Two new references on the topic have been added.

6) On page 11 the authors mention their rate equations in Section S8 and emphasize that they are not sufficient for describing the condensate or lasing observations. This raises the question how the rate equations would have to be extended to cover at least qualitatively all the experimentally observed features.

The rate equations in Supplementary Note 5 (Section S8 in the previous version) are exceedingly simple, and their only purpose is to describe a general stimulated pulse formation process, in order to explain

the low intensity areas in Fig.5 and thereby to reveal the stimulated nature of the redshift process. To describe condensation or lasing phenomena, one should not try to extend these equations but rather take a more microscopic starting point. A good starting point would be the Hamiltonian from which the Kirton-Keeling model has been derived, however, to describe the strong coupling regime where we work, one cannot do perturbation theory in terms of the light-matter interaction as in the original Kirton-Keeling model (please see the item below). These issues should now become more clear from the new paragraphs (last three of the Section "Stimulated nature of the thermalization") that we have added to the manuscript.

*7) The authors mention on page 13 that the Kirton-Keeling dynamics in Section S9 is not sufficient to describe the observed quantum dynamics. And they attribute this to the high coherence of the thermalization due to the observed stimulated processes and due to strong light-matter coupling. Therefore I suggest that the authors apply an extension of the Kirton-Keeling dynamics to their system which takes into account self-consistently not only the dissipative but also the coherent dynamics:
New J. Phys. 20, 055014 (2018)*

The calculations in Supplementary Figure 5 and Supplementary Note 6 (previously Section S9) have the same microscopic starting point as the Kirton-Keeling model, but in contrast to that model, we do not assume weak coupling between the light and the molecules. The Kirton-Keeling model uses weak coupling approximation to do perturbation theory, allowing to treat systems of realistic size. The reason why we can do calculations without the weak coupling approximation is that we choose a very small system: just one molecule, one vibrational state, and a few light modes. Supplementary Note 6 thus presents a toy-model and the results may give qualitative ideas of what happens in a larger system.

We thank the reviewer for the suggestion to apply the method presented in New J. Phys. 20, 055014 (2018). We have previously applied the non-equilibrium model of photon condensation by Kirton and Keeling (Manuscript Ref. (53)) to describe our weak coupling experiments (Manuscript Ref. (28)). However, for the current experiments one needs a model that is valid under strong coupling. Therefore, even though the model presented in New J. Phys. 20, 055014 (2018) includes coherent terms that were omitted in the original model by Kirton and Keeling, it still assumes weak coupling such that the light-emitter interaction can be treated under the perturbation theory. To model a strongly coupled system, the interaction term must be kept as part of the Hamiltonian (as we do in Equation (9) of Supplementary Note 6). We have added the reference New J. Phys. 20, 055014 (2018) to page 16, where we refer to the current state-of-the-art theory, because it is useful to know for anyone following the line of work for photon condensate models.

8) At the end of their manuscript the authors discuss the effect of pulse duration. Here I suggest to make the manuscript more concise by producing and discussing a phase diagram, where the pump fluence for the occurrence of the first and second threshold are plotted versus the pump duration. Such a phase diagram would summarize within one figure various experimental results, which are currently distributed in different figures.

We thank the referee for the suggestion to present the results of the effect of pulse duration in this way. We have added the new Figure 7c which presents the thresholds for lasing and condensation, as well as the start of the incomplete stimulated thermalization regime, as a function of pulse duration. One can nicely see that while the lasing threshold increases only modestly for increasing pulse duration, for the thermalization and condensation regimes the increase is more rapid.

In addition to changes answering the reviewers' comments, we have done format changes to follow the journal style (much of previous Supplementary information is in Methods now, for instance).

We thank the reviewer once more and hope our responses and the revisions to the manuscript are satisfactory and the manuscript is now ready for publication in Nature Communications.

Reviewers' comments:

Reviewer #1 (Remarks to the Author):

Dear editor and authors,

the resubmitted manuscript by Väkeväinen et al. has significantly improved, and the conclusions are now much better backed up by the measured data. Before I can recommend publication, there are however a few points that need to be addressed:

i) Just for my own curiosity: How stable are the real space images in fig. 2? If instead of single shot (in b, d and f) one would average over the same amount of shots as for the spectral data in c, e and g, which of the observed features do survive?

ii) In Fig. 3 h and i, one should in principle see a thermal tail, however in the present color scale this is just some shadow. Could one make this tail observable by e.g. using a logarithmic color scale, or does the blooming of the CCD somehow prevent this? Considering that the integrated signal does not suffer severely from the blooming, I guess this should be an option?

iii) As a suggestion, the authors might want to discuss the nature of the "multimode condensation" in a little more detail. It might indeed be a single mode condensate at any given time, but due to the blue shift which is a function of intensity and thus also of time due to the pump pulse temporal profile the condensate mode will change. Then the observed spectra would be somehow "dynamical single mode" instead of multimode. I am aware that temporal resolution is out-of-the-way, but maybe there are other signatures that could help decide between the two options?

iv) On page 12, the authors give the scale of $25\mu\text{m}$ after which the red shift occurs. With this part, I have a few points that should be addressed:

- The $25\mu\text{m}$ is presented as something general, however it is only true for a given pump fluence. While this is clear from fig. 5i, it should be clearly addressed in the text also.
- The authors should explain how they experimentally determine, at which point the red shift starts. I got the feeling that this is done "by eye" when reading the text, but I assume there is a more clear-cut criterion for that?
- The authors should discuss the existence of the saturation value of $18\mu\text{m}$ in more detail. It is unclear to me where it should come from, and if/how it could be at least estimated from system parameters? The $18\mu\text{m}$ corresponds roughly to the distance a free space would travel during the pulse duration of 50 fs, but this is clearly a weird coincidence. However, it is quite striking that the Buildup-Time model from Q-switching fits so nicely but then suddenly breaks down, thus this needs an explanation.

v) In figure 5, integrated spectra would be quite convenient such that one can estimate the degree of thermalization "by eye". According to Fig. 2, the pump fluence of $2.2\mu\text{J}$ is rather close to the condensation threshold, thus I believe this would be quite helpful (I think one of them corresponds to something slightly above the yellow line in Fig. S6, but a single, well readable spectrum in the supplementary for each lattice size would be quite helpful).

vi) On page 17, the authors state "It seems that for a long excitation pulse...[]...as it competes from the same gain with the lasing...". However, the competition for gain is always the case, independent of the pump pulse length, so I do not quite get the intention of that statement. It does not really explain the difference between short and long pulses.

vii) In fig. 7, the authors state that they chose the threshold for thermalization (i.e. the data points for incomplete thermalization) from the point at which the interference patterns become visible. This seems a completely subjective measure, as "something becoming visible" obviously is a matter of color scale, noise, personal taste, and such like. Correspondingly, for these data points

I would ask for a more reliable measure such as "the visibility of the associated interference pattern is larger than α ", which is independent on the observer.

viii) In their reply letter the authors state that in light of my previous comment (Reviewer 1, statement vii) they changed the conclusion to "These questions have become challenging for strongly coupled room temperature condensates", as most of the claimed points were discussed for the Photon BECs in Bonn and London. However, the first sentence of the discussion still gives the impression that nothing has been done in that direction. I would appreciate a reformulation in the sense "These questions have been addressed for weakly coupled photon BECs, but become challenging for strongly coupled systems..." or something in this spirit.

Reviewer #2 (Remarks to the Author):

I have read the comments of all referees, the replies by the authors, and the revised manuscript & supplemental material in detail. The authors have replied in a lot of detail to all referee comments and questions and as a result have significantly improved the manuscript. I believe all important questions have been addressed and thus now recommend publication in Nature Communications without further revisions.

Reviewer #3 (Remarks to the Author):

During the revision process the manuscript has been considerably improved along the lines of the criticism of the referees. This has clarified the majority of points, which have been raised. But, partially due to the enormous scope of both the manuscript and the supplemental material, still some points are unclear to me, so I hope that the refereeing process of Nature Communications would allow also their clarification:

1) The authors answered quite elaborately concerning the question whether they observe a Bose-Einstein or a Maxwell-Boltzmann distribution. Partially I do not understand their extensive argumentation. Somehow I would have expected that they plainly plot the experimental data in form of $\ln(1+1/n)$ as a function of energy E . In case of a Bose-Einstein distribution $\ln(1+1/n)$ would coincide with $(E-\mu)/(k_B T)$. Thus, the slope should yield the temperature and the axis section the chemical potential. As a consequence, the question is how well the temperature in the experiment is reproduced and how the result for the chemical potential could be understood in terms of the ground-state energy of the system.

2) In view of the photon-photon interaction strength, the authors mention now a value of about $0.2 \mu e V \mu m^{-2}$. Here it would be helpful if the authors could also calculate the dimensionless interaction strength $\tilde{g} = g m / \hbar^2$, which is possible to determine due to the two-dimensionality of the considered system. To this end one would have to know the polariton mass in the experiment. The resulting value would be important to judge whether the polaritons are in the state of a BEC, i.e. $\tilde{g} < 0.5$, or in a BKT, i.e. $\tilde{g} > 0.5$. Please note that this boundary value 0.5 is not a sharp one but just gives the order of magnitude.

3) The plots in Fig. 4 are unclear to me. On the one hand, the contrast is defined in the method section in Eq. (3) locally, on the other hand the plots in Fig. 4 do not contain any spatial dependence. It is just mentioned that the contrast is plotted over the region of interest – but what does this mean concretely? Does a point in Fig. 4 correspond to the integral of the contrast in the region of interest

or is it the average or just the contrast value evaluated at half the size of the region of interest? Here a clarification would be appreciated by the readers.

After the clarification of these points, the manuscript could be published in the journal Nature Communications.

Response to the Reviewer reports

Sub-picosecond thermalization dynamics in condensation of strongly coupled lattice plasmons

Aaro I. Väkeväinen, Antti J. Moilanen, Marek Nečada,
Tommi K. Hakala, Konstantinos S. Daskalakis, Päivi Törmä

Reviewer: 1

Dear editor and authors, the resubmitted manuscript by Väkeväinen et al. has significantly improved, and the conclusions are now much better backed up by the measured data. Before I can recommend publication, there are however a few points that need to be addressed:

i) Just for my own curiosity: How stable are the real space images in fig. 2? If instead of single shot (in b, d and f) one would average over the same amount of shots as for the spectral data in c, e and g, which of the observed features do survive?

Throughout our studies, we have done comparisons of images recorded for single pump pulses and those averaged over many pump pulses. The features stay qualitatively the same.

ii) In Fig. 3 h and i, one should in principle see a thermal tail, however in the present color scale this is just some shadow. Could one make this tail observable by e.g using a logarithmic color scale, or does the blooming of the CCD somehow prevent this? Considering that the integrated signal does not suffer to severely from the blooming, I guess this should be an option?

We have tried the logarithmic scale and although it enhances the visual appearance of the tail, the blooming artifacts become so dominant that it is disturbing rather than beneficial for the overall quality of the pictures. Furthermore the dispersion plots are anyway presented in false color, and the quantitative result of integrated intensity distribution is presented in the line spectra in log-scale. For these reasons we have decided to keep the color plots as they are (in linear scale). Please note that the chosen false color map already enhances the features in the low end of the intensity scale (light blue). The chosen color map works well in the computer screen. We admit that the image quality suffers if printed out on paper.

The Figs. 3h and i display results integrated over a number of pump pulses. This information was missing from the figure caption, it has now been added. The line spectrum Fig. 3j is integrated over k_y (as was already mentioned in the figure caption), and indeed it looks good in logarithmic scale and the lowest panel shows a clear thermal tail.

iii) As a suggestion, the authors might want to discuss the nature of the "multimode condensation" in a little more detail. It might indeed be a single mode condensate at any given time, but due to the blue shift which is a function of intensity and thus also of time due to the pump pulse temporal profile the condensate mode will change. Then the observed spectra would be somehow "dynamical single mode" instead of multimode. I am aware that temporal resolution is out-of-the way, but maybe there are other signatures that could help decide between the two options?

This is a very good point, we agree with the referee that such time-evolving condensation is possible. We have added the sentence "Since we observe a temporally integrated signal, we cannot rule out the possibility of a single mode condensate temporally evolving between different states in the sub-picosecond

scale.” to the manuscript on page 9, in the end of the paragraph where the multiple peaks are discussed, to point out to the reader that the multimode vs. single mode question needs further study.

Concerning signatures, most theory works associate single mode condensation with thermal equilibrium, so the prominent thermal tail that we observe might hint towards single mode condensation. But theory that properly describes thermalization in the strong coupling regime would be needed to back up such claim (and such theory does not exist at the moment, as explained in the manuscript). Another possible signature is that, in case the condensate population oscillates, high-resolution temporal coherence measurements would show a sideband. If the time evolution is not oscillatory, it might nevertheless lead to some other features, such as broadening. We thank the referee for bringing our attention to this point and will keep it in mind in our future research.

iv) On page 12, the authors give the scale of 25 μm after which the red shift occurs. With this part, I have a few points that should be addressed:

- The 25 μm is presented as something general, however it is only true for a given pump fluence. While this is clear from fig. 5i, it should be clearly addressed in the text also.

Thanks for pointing this out, we have changed the sentence in the parenthesis (page 12, end of second paragraph) to highlight that the results shown in Fig 5a,c,e are for a given pump fluence: “ $\sim 25\mu\text{m}$ for 2.2 mJcm^{-2} pump fluence presented in Figure 5a,c,e”.

- The authors should explain how they experimentally determine, at which point the red shift starts. I got the feeling that this is done “by eye” when reading the text, but I assume there is a more clear-cut criterion for that?

This important information was missing from the manuscript. We have utilized cross sections of integrated intensity over the x-axis of the real space spectra. We have determined the starting point of red shift to equal the rising edge of the intensity with respect to the array edge. How the location where the red shift begins is determined is now explained by a new Supplementary Figure 10. We refer to this Supplementary Figure on page 12 when discussing the starting point of the red shift in the main text.

- The authors should discuss the existence of the saturation value of 18 μm in more detail. It is unclear to me where it should come from, and if/how it could be at least estimated from system parameters? The 18 μm correspond roughly to the distance a free space would travel during the pulse duration of 50 fs, but this is clearly a weird coincidence. However, it is quite striking that the Buildup-Time model from Q-switching fits so nicely but then suddenly breaks down, thus this needs an explanation.

The saturation happens at a pump fluence that corresponds to the transition to the BEC. Apparently, the simple pulse build-up model describes reasonably well the size of the low-intensity area in the incomplete thermalization regime, but not in the BEC one. Why this is so we do not know, but it is an extremely interesting question which we hope to answer in the future. We have now changed the sentence “Figure 5i shows that the dark-zone width follows the inverse of the pump fluence until it saturates at around 3 mJcm^{-2} to a value below 20 μm ($\sim 100 - 140$ fs).” into the form “Figure 5i shows that the dark-zone width follows the inverse of the pump fluence until it saturates at around 3 mJcm^{-2} (corresponding to the BEC threshold) to a value below 20 μm ($\sim 100 - 140$ fs).” This tells that the saturation is associated with the BEC transition.

v) In figure 5, integrated spectra would be quite convenient such that one can estimate the degree of thermalization “by eye”. According to Fig. 2, the pump fluence of 2.2uJ is rather close to the condensation threshold, thus I believe this would be quite helpful (I think one of them corresponds to something slightly

above the yellow line in Fig. S6, but a single, well readable spectrum in the supplementary for each lattice size would be quite helpful).

We have added a new Supplementary Figure 11 presenting the integrated spectra for the three different lattice sizes at pump fluence 2.2 mJcm^{-2} corresponding to manuscript Figure 5. We mention this new figure in the figure caption of Fig. 5 of the main text.

vi) On page 17, the authors state "It seems that for a long excitation pulse...[.]...as it competes from the same gain with the lasing...". However, the competition for gain is always the case, independent of the pump pulse length, so I do not quite get the intention of that statement. It does not really explain the difference between short and long pulses.

Yes, this sentence was written in a misleading way; the competition is always there, independent of the pulse length. We have reformulated the text to read: "For all pulse lengths, the thermalization process competes from the same gain with the lasing. It seems that for a long excitation pulse the instantaneous population inversion does not reach a high enough value for the thermalization process to take over the lasing which is already triggered at the first threshold (see also Figure 6)."

vii) In fig. 7, the authors state that they chose the threshold for thermalization (i.e. the data points for incomplete thermalization) from the point at which the interference patterns become visible. This seems a completely subjective measure, as "soemthing becoming visible" obviously is a matter of color scale, noise, personal taste, and such like. Correspondingly, for these data points I would ask for a more reliabel measure such as "the visibility of the associated interference pattern is larger than xx ", which is independent on the observer.

We are grateful for the reviewer pointing this out, because the comment about visibility was misleading. We didn't use visibility as a criterion, instead, we determined the threshold for thermalization by following the evolution of maxima in the integrated line spectra as function of the pump fluence. In the lasing regime, there is one maximum corresponding to the band edge (this is the lasing peak), and another one (smaller in intensity) at higher energy. First, for increasing pump fluence, the lasing peak grows and the higher energy peak diminishes. At a certain pump fluence, however, the higher energy peak starts growing, with respect to the lasing peak. We determine this pump fluence value as the threshold for the incomplete thermalization regime. It can be easily and uniquely determined from the line spectra. It happens to coincide with pump fluence values where the fringes become visible (just by visual inspection of the data), this explains our misleading sentence about the threshold. We now give the actual definition of threshold in the caption of Fig. 7, and then just mention that at this fluence the fringes can be typically seen in the data.

viii) In their reply letter the authors state that in light of my previous comment (Reviewer 1, statement vii) they changed the conclusion to "These questions have become challenging for strongly coupled room temperature condensates", as most of the claimed points were discussed for the Photon BECs in Bonn and London. However, the first sentence of the discussion still gives the impression that nothing has been done in that direction. I would appreciate a reformulation in the sense "These questions have been adressed for weakly coupled photon BECs, but become challenging for strongly coupled systems..." or something in this spirit.

We agree. We have changed the sentence to the form suggested by the reviewer: "These questions have been addressed for weakly coupled BECs, but become challenging to answer for strongly coupled room temperature condensates as higher energy scales imply faster dynamics."

We thank the reviewer for reading of our revised manuscript and response, and for the above comments

that have helped to clarify some remaining essential points. We hope the manuscript can now be published in Nature Communications.

Reviewer: 2

I have read the comments of all referees, the replies by the authors, and the revised manuscript & supplemental material in detail. The authors have replied in a lot of detail to all referee comments and questions and as a result have significantly improved the manuscript. I believe all important questions have been addressed and thus now recommend publication in Nature Communications without further revisions.

Reviewer: 3

During the revision process the manuscript has been considerably improved along the lines of the criticism of the referees. This has clarified the majority of points, which have been raised. But, partially due to the enormous scope of both the manuscript and the supplemental material, still some points are unclear to me, so I hope that the refereeing process of Nature Communications would allow also their clarification:

1) The authors answered quite elaborately concerning the question whether they observe a Bose-Einstein or a Maxwell-Boltzmann distribution. Partially I do not understand their extensive argumentation. Somehow I would have expected that they plainly plot the experimental data in form of $\ln(1 + 1/n)$ as a function of energy E . In case of a Bose-Einstein distribution $\ln(1 + 1/n)$ would coincide with $(E - \mu)/(k_B T)$. Thus, the slope should yield the temperature and the axis section the chemical potential. As a consequence, the question is how well the temperature in the experiment is reproduced and how the result for the chemical potential could be understood in terms of the ground-state energy of the system.

We thank the referee for the excellent suggestion to represent our data as $\ln(1 + 1/n)$, and obtaining the temperature and chemical potential from such plots. Analyzing the distributions as $\ln(1 + 1/n)$ requires that the distribution $n = 1/(\exp((E - \mu)/k_B T) - 1)$. In a general case, the observed distribution may have a degeneracy factor that can depend on energy, $g(E)$, and the observed photon number may differ from the one in the condensate due to losses in the detection path (let us denote this suppression factor by A). The observed distribution is then $n = Ag(E)/(\exp((E - \mu)/k_B T) - 1)$, and $\ln(1 + 1/n)$ no longer gives $(E - \mu)/(k_B T)$. One can of course divide the data by A and $g(E)$ to get a scaled \tilde{n} , if they are known, and then use the $\ln(1 + 1/\tilde{n})$ plot. We can utilize the $g(E)$ that we already use in our fits, but we know the factor A only at the order of magnitude level, not its precise value (please see section "Estimation of photon number in the condensate" in Methods; it contains many "about" and "roughly" estimates). Thus plotting $\ln(1 + 1/n)$ is not useful in our case. We have checked, however, that the $\ln(1 + 1/n)$ plot gives roughly the expected μ (the energy at which the condensation happens).

In contrast, fitting the thermal tail to the Maxwell-Boltzmann distribution does not require the absolute photon number (i.e., precise knowledge of the factor A), therefore in our case this is the better option.

2) In view of the photon-photon interaction strength, the authors mention now a value of about $0.2\mu\text{eV}\mu\text{m}^2$. Here it would be helpful if the authors could also calculate the dimensionless interaction strength $\tilde{g} = gm/\hbar^2$, which is possible to determine due to the two-dimensionality of the considered system. To this end one would have to know the polariton mass in the experiment. The resulting value would be important to judge whether the polaritons are in the state of a BEC, i.e. $\tilde{g} < 0.5$, or in a BKT, i.e. $\tilde{g} > 0.5$. Please note that this boundary value 0.5 is not a sharp one but just gives the order of magnitude.

We have calculated the dimensionless interaction strength using the previously calculated mean-field

approximation estimate $g = 0.2\mu\text{eV}\mu\text{m}^2$ and a polariton mass obtained by fitting a parabola into the band edge of the lower polariton dispersion branch. Fitting to both TM and TE modes of the coupled system gives an effective mass in range $10^{-37}\dots 10^{-35}$ kg. Using these values and the calculated g , the dimensionless interaction strength $\tilde{g} = gm/\hbar^2$ is of the order of $10^{-7}\dots 10^{-5}$. This is of similar magnitude as reported e.g. in microcavity photon condensates and indicates that the polaritons correspond to a BEC rather than BKT.

We have added a new Supplementary Note 8: Estimation of polariton-polariton interaction strength, where we present the calculated numbers and cite papers relevant for the calculation: Bloch et al., Rev. Mod. Phys. 80, 885 (2008), Deng et al. Rev. Mod. Phys. 82, 1489 (2010), Carusotto et al., Rev. Mod. Phys. 85, 299 (2013) and Radonjic et al., New J. Phys. 20, 055014 (2018).

3) The plots in Fig. 4 are unclear to me. On the one hand, the contrast is defined in the method section in Eq. (3) locally, on the other hand the plots in Fig. 4 do not contain any spatial dependence. It is just mentioned that the contrast is plotted over the region of interest – but what does this mean concretely? Does a point in Fig. 4 correspond to the integral of the contrast in the region of interest or is it the average or just the contrast value evaluated at half the size of the region of interest? Here a clarification would be appreciated by the readers.

Yes, the explanation of the spatial coherence measurement should be improved. The reported contrast values in Fig. 4a are the average contrast values obtained from the region of interest, which is where the condensate has high intensity. We do not investigate the spatial coherence as a function location as we are interested in how it evolves (on average) as a function of pump fluence. The evolution of spatial coherence allows us to clearly distinguish the three different regimes (lasing, incomplete thermalization, condensation) as a function of pump fluence. We now tell in the caption of Fig. 4 that 4a) presents the contrast averaged over the region of interest.

After the clarification of these points, the manuscript could be published in the journal Nature Communications.

We thank the reviewer for reading all the resubmission material and for recommending publication of our manuscript in Nature Communications, provided the above points are clarified. We hope the clarifications have been satisfactorily done and the manuscript is now ready for publication in Nature Communications.

REVIEWERS' COMMENTS:

Reviewer #1 (Remarks to the Author):

Dear editor and authors,

my concerns have been addressed, I am happy to recommend the paper for publication in Nature Communications.

Reviewer #2 (Remarks to the Author):

I already supported publication in the previous round, and after reviewing the responses to the comments of the other referees, I believe the paper is even stronger now, and thus even more support publication in Nature Communications.

Reviewer #3 (Remarks to the Author):

As a result of the second revision process the authors have further refined both the manuscript and the supplemental material by incorporating the criticisms of the referees point by point. This has been accomplished to such a level of detail that now the manuscript is ready for being published at the journal „Nature Communications“.